# Characterization of histone acylations links chromatin modifications with metabolism

Johayra Simithy[1], Simone Sidoli[1], Zuo-Fei Yuan [1], Mariel Coradin[1], Natarajan V. Bhanu[1], Dylan M. Marchione[2], Brianna J. Klein[3], Gleb A. Bazilevsky [4], Cheryl E. McCullough[5], Robert S. Magin[6], Tatiana G. Kutateladze[3], Nathaniel W. Snyder[7], Ronen Marmorstein [8] & Benjamin A. Garcia[1]

Over the last decade, numerous histone acyl post-translational modifications (acyl-PTMs) have been discovered, of which the functional significance is still under intense study. Here, we use high-resolution mass spectrometry to accurately quantify eight acyl-PTMs in vivo and after in vitro enzymatic assays. We assess the ability of seven histone acetyltransferases (HATs) to catalyze acylations on histones in vitro using short-chain acyl-CoA donors, proving that they are less efficient towards larger acyl-CoAs. We also observe that acyl-CoAs can acylate histones through non-enzymatic mechanisms. Using integrated metabolomic and proteomic approaches, we achieve high correlation ($R^2 > 0.99$) between the abundance of acyl-CoAs and their corresponding acyl-PTMs. Moreover, we observe a dose-dependent increase in histone acyl-PTM abundances in response to acyl-CoA supplementation in in nucleo reactions. This study represents a comprehensive profiling of scarcely investigated low-abundance histone marks, revealing that concentrations of acyl-CoAs affect histone acyl-PTM abundances by both enzymatic and non-enzymatic mechanisms.

[1] Department of Biochemistry and Biophysics, Perelman School of Medicine, University of Pennsylvania, Philadelphia, PA 19104, USA. [2] Department of Systems Pharmacology and Translational Therapeutics, Perelman School of Medicine, University of Pennsylvania, Philadelphia, PA 19104, USA. [3] Department of Pharmacology, University of Colorado School of Medicine, Aurora, CO 80045, USA. [4] Graduate Group in Cell and Molecular Biology, Perelman School of Medicine, University of Pennsylvania, Philadelphia, PA 19104, USA. [5] Department of Chemistry, University of Pennsylvania, Philadelphia, PA 19104, USA. [6] Graduate Group in Biochemistry and Molecular Biophysics, Perelman School of Medicine, University of Pennsylvania, Philadelphia, PA 19104, USA. [7] AJ Drexel Autism Institute, Drexel University, 3020 Market Street Suite 560, Philadelphia, PA 19104, USA. [8] Department of Biochemistry and Biophysics, Abramson Family Cancer Research Institute, and the Department of Chemistry, University of Pennsylvania, Philadelphia, PA 19104, USA. Correspondence and requests for materials should be addressed to B.A.G. (email: bgarci@mail.med.upenn.edu)

Lysine acetylation is the most extensively studied histone post-translational modification (PTM). Discovered more than 50 years ago, it has been recognized to play a fundamental role in transcriptional activation, metabolic regulation and other central cellular processes[1]. Mechanistically, lysine acetylation neutralizes the positive charge of histone tails, reducing the physical interaction between histones and DNA, thereby allowing the access of gene-activating transcription factors[2,3]. Acetylation can also influence chromatin function by serving as a binding site for bromodomain-containing remodeling complexes that can directly stimulate trasncription by recruiting transcriptional co-activators[4]. Acetylation dynamics in the nucleosome are the result of the net activities between two different families of enzymes: histone acetyltransferases (HATs), which catalyze the

addition of acetyl groups using acetyl-CoA as a cofactor, and histone deacetylases (HDACs), which remove these groups[5]. The activities of both HATs and HDACs are regulated by the metabolic state of the cell[6]. Thus, endogenous metabolite concentrations are proposed to provide signaling that can directly influence acetylation dynamics in chromatin[7].

Over the last decade, a growing number of lysine modifications chemically related to acetylation (propionylation, malonylation, crotonylation, butyrylation, succinylation, glutarylation, 2-hydroxyisobutyrylation and β-hydroxybutyrylation) have been identified on histones using mass spectrometry (MS)-based proteomic approaches[8–14]. These findings have raised numerous questions regarding their functional significance, possible implications in metabolic pathways and the existence of

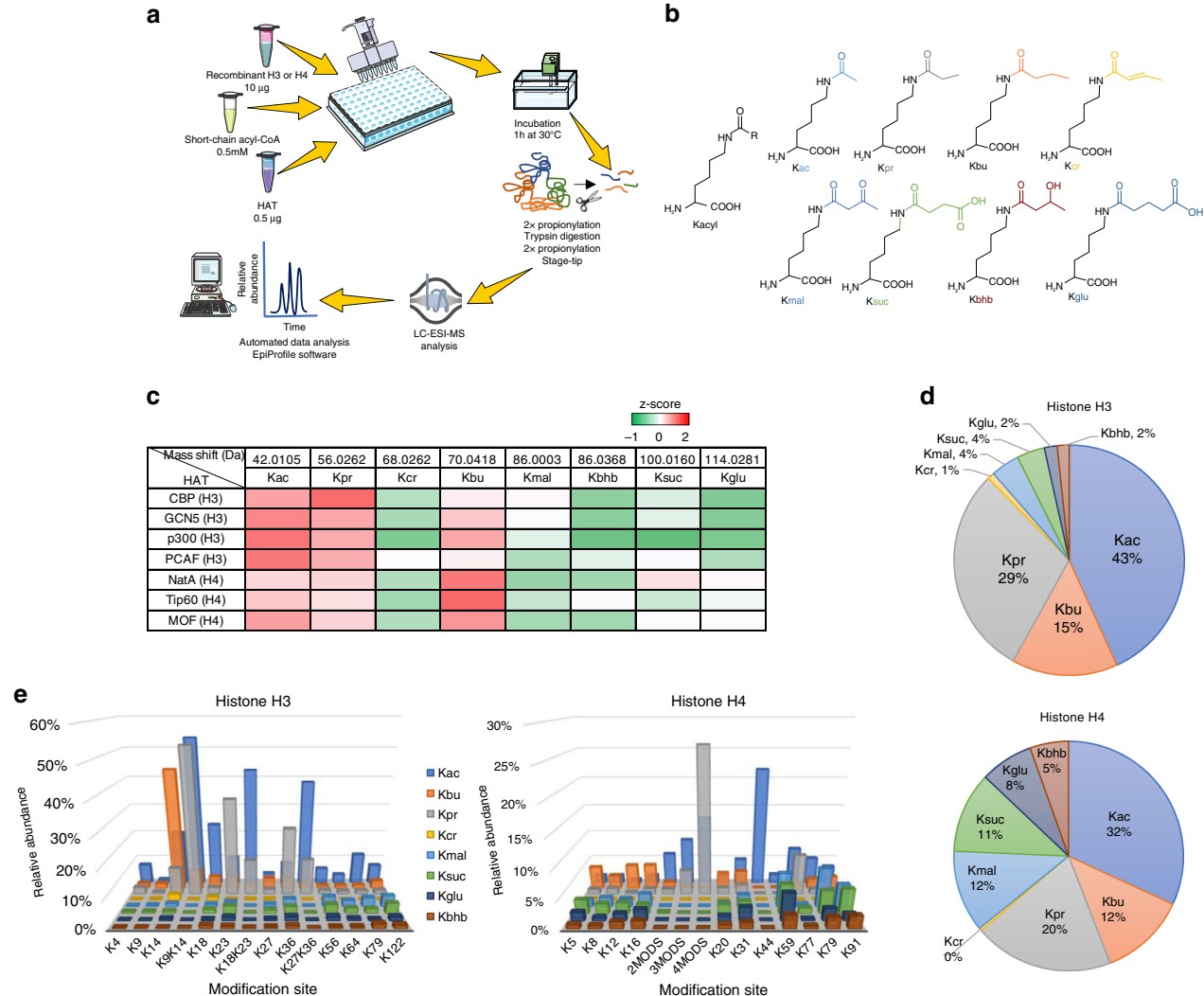

**Fig. 1** Overview of histone acetyltransferases (HATs) in vitro acylation activity and specificity. **a** Schematic representation of in vitro acylation assay. **b** Chemical structures of histone acyl modifications evaluated in this study. Lysine modifications and abbreviations are: acetyl (Kac), propionyl (Kpr), butyryl (Kbu), crotonyl (Kcr), malonyl (Kmal), succinyl (Ksuc), β-hydroxybutyryl (Kbhb), and glutaryl (Kglu). **c** Heat map displaying the in vitro acylation activity profiles of different HATs in the presence of acetyl-, propionyl-, crotonyl-, butyryl-, malonyl-, β-hydroxybutyryl-, succinyl- and glutaryl-CoA. Molecular mass shift of the various acylated lysines residues are shown in the table headers. Different HATs were assayed against histones H3 or H4 as specified in the first column. To generate the heat map, we averaged the relative abundance of acyl-PTMs on the quantified peptides and then normalized (z-scores) those values across the different HATs, i.e. row normalization. **d** Pie chart showing the average relative frequency of in vitro acylated peptides, divided in results for histone H3 (top) and histone H4 (bottom). **e** Bar plots depict the specificity for all HATs on the histone sequence. The x axis represents the modification site at histones H3 (left) and histone H4 (right), and the y axis represents the relative abundance shown as the average contribution of HATs to all acylated peptides. For histone H4 N-terminal peptide (G4-R17), the number of acylations on the sequence are displayed using the code 2, 3 or 4 mods. This is because it was not always possible to discriminate modification sites on the multiply modified H4 peptide. All values shown were corrected by the contribution of non-enzymatic acylation. All results are shown as the average of 3 independent experiments

regulatory enzymes beyond the well-established acetylation mechanisms that could govern these marks. While most of these questions remain to be answered, many studies have provided new insights into the roles that acyl marks can play in genome function. For example, it has been reported that lysine crotonylation mediated by the HAT p300 can stimulate gene transcription in vitro and in vivo seemingly to a greater degree than lysine acetylation, and that this mechanism is highly regulated by the metabolic concentrations of the crotonyl-CoA co-factor[15]. Histone crotonylation has also been found to be enriched at transcriptionally active X/Y sex-linked genes during post-meiotic sex inactivation in mouse[8,16], and the YEATS domains Taf14, AF9 and YEATS2 have been reported to preferentially bind crotonylated over acetylated lysines residues in vitro[17–20]. More recently, lysine butyrylation has been reported to directly stimulate gene transcription and compete with acetylation for the binding of the testis specific gene expression-driver Brdt in spermatogenic cells[21]. In addition, β-hydroxybutyrylation, was found to be induced during starvation or streptozotocin-induced diabetic ketoacidosis, and to activate transcription of specific genes associated with starvation-responsive metabolic pathways[14]. These studies suggest that newly identified histone acyl-PTMs may have unique or similar roles to acetylation in transcriptional activation. Such observations are supported by several reports showing that SIRT5, a member of the class III HDACs can preferentially remove acidic acyl modifications, including malonyl[22], succinyl[22,23] and glutaryl[13], whereas propionyl, crotonyl and butyryl marks can be removed by various other sirtuins[24]. However, it remains unclear whether the same group of enzymes involved in the establishment of acetylation could also mediate the establishment of these histone modifications in vivo.

Another aspect that has been underexplored is the relative abundance of acyl marks, which is an important step towards understanding their biological relevance. This gap in knowledge is mainly due to the biases inherent in the use of antibody-based enrichment methods commonly employed prior to MS detection[25]. Although stoichiometry at individual sites has been reported for propionylation (7%) at H3K23 in a leukemia cell line[26], butyrylation (31%) at H3K115 in mouse brain[27], and crotonylation (1–3%) at H2AK36, H2BK5, H3K23 and H4K12 in brain histones[27], a global overview of the abundances relative to histone acetylation is lacking. The dearth of quantitative data for these non-canonical acyl-PTMs has led to the hypothesis that they might arise due to the chemical reactivity of acyl-CoAs. Indeed, this has been observed in the context of acetylation and succinylation in mitochondrial proteins[28]. Thus, it is still open to debate whether these modifications are strategically positioned on the chromatin by enzymes or whether their presence is instead the result of non-enzymatic chemical reactivity.

In this study, we sought to characterize the acylation of histones, including their overall abundance and their likelihood to be products of HAT catalysis rather than chemical reactivity of acyl-CoAs. First, we investigate the ability of several recombinant HATs to catalyze the acylation of histones H3 and H4 using different acyl-CoA donors employing an MS-based in vitro bioassay. Our data show that most HATs can catalyze histone acylation using different acyl-CoA substrates to variable extents when tested individually. However, in competition assays performed in the presence of equimolar concentrations of acyl-CoA and acetyl-CoA, almost all HATs strongly prefer to utilize acetyl-CoA to modify histones. Our data also confirm that histones can be modified non-enzymatically through the chemical reactivity of the different acyl-CoA donors alone. We also employ a proteomics approach to characterize several acyl-PTMs in nucleo and in vivo without the use of enrichment strategies, which allow us to determine the relative abundance of a diverse subset of acylations on histones. Because these marks are dependent on their corresponding short-chain acyl-CoA metabolic intermediates, we employ a targeted metabolomics approach to measure the concentrations of acyl-CoA metabolites in HeLa cells and in proliferative and differentiated human myogenic cells. We find that the cellular concentrations of different acyl-CoA metabolites span orders of magnitude and are tightly correlated with the relative abundances of the acyl marks identified in vivo. These findings support the notion of a direct link between cellular metabolism and epigenetic regulation, where the relative abundances of different acyl marks in histones are driven by the cellular concentrations of their respective metabolic intermediates[29,30].

## Results

**In vitro acylation of histones H3 and H4**. Previous studies have reported that the HATs CBP, p300 and PCAF can mediate propionylation[12], butyrylation[12,31], and crotonylation[15] of lysine residues in vitro. These observations prompted us to investigate whether these and other known HATs can catalyze the acylation of human recombinant histones H3 and H4 using a broader range of acyl-CoA donors. We performed in vitro HAT activity assays with the HAT domains of PCAF, Gcn5, and the full-length CBP and p300 enzymes against histone H3, and with the HAT domains of MOF, Tip60 and NatA against histone H4. Each reaction was carried out individually in the presence of eight different short-chain acyl-CoA donors, followed by bottom-up nano-LC-MS/MS analysis (Fig. 1a, b). Figure 1c summarizes the in vitro activity profiles of all HATs evaluated in this study. The heat map shows that most HATs could utilize acetyl-propionyl- and butyryl-CoA with relatively high efficiency, supporting recent findings[32,33]. However, acidic acyl-CoA donors including malonyl-, succinyl- and glutaryl-CoA, and branched-chain acyl donors like β-hydroxybutyryl CoA are utilized by HATs less efficiently. Interestingly, enzymes did not seem to utilize crotonyl-CoA for the catalysis of acyl marks as effectively as propionyl- and butyryl-CoA despite the structural similarity within these cofactors. These data are in agreement with previous observations suggesting that HATs activity is weaker with crotonyl-CoA due to the planarity and rigidity imparted by the C-C double bond in the crotonyl moiety[8,20,32].

When taking a closer look at the individual acylation activities of all HATs (Supplementary Table 1), we observed that enzymes have different trends in their substrate preference. For instance, histone H3 is known to be selectively acylated at the lysine residue 14 (H3K14) by Gcn5[33] and PCAF[34]. Therefore, if we look at the relative abundances for all acylations at H3K14, under our assay conditions, Gcn5 was able to butyrylate ~ 78% of the H3 peptide at position 14, whereas acetylation and propionylation were found at ~ 32% and ~ 11%, respectively (Supplementary Table 1). Likewise, PCAF displayed ~ 88% butyrylation, followed by ~ 5% crotonylation and ~ 2% acetylation at position H3K14. However, when looking at the average sum of the relative abundances of all acylated peptides by HATs on H3, there seems to be a trend in the order of substrate preference: acetyl > propionyl > butyryl > malonyl > succinyl > β-hydroxybutyryl > glutaryl > crotonyl (Fig. 1d; Supplementary Table 1). This trend inversely correlates with the increasing size of the side chain of the acyl donor (except for crotonyl-CoA and β-hydroxybutyryl), supporting the notion that the activity of HATs gets weaker with increasing acyl-chain length[32]. Interestingly, p300 and PCAF were the enzymes with the highest crotonylation activities on H3 (Supplementary Table 1).

Moreover, the average activities of HATs on histone H4 showed patterns that were consistent with the trend mentioned

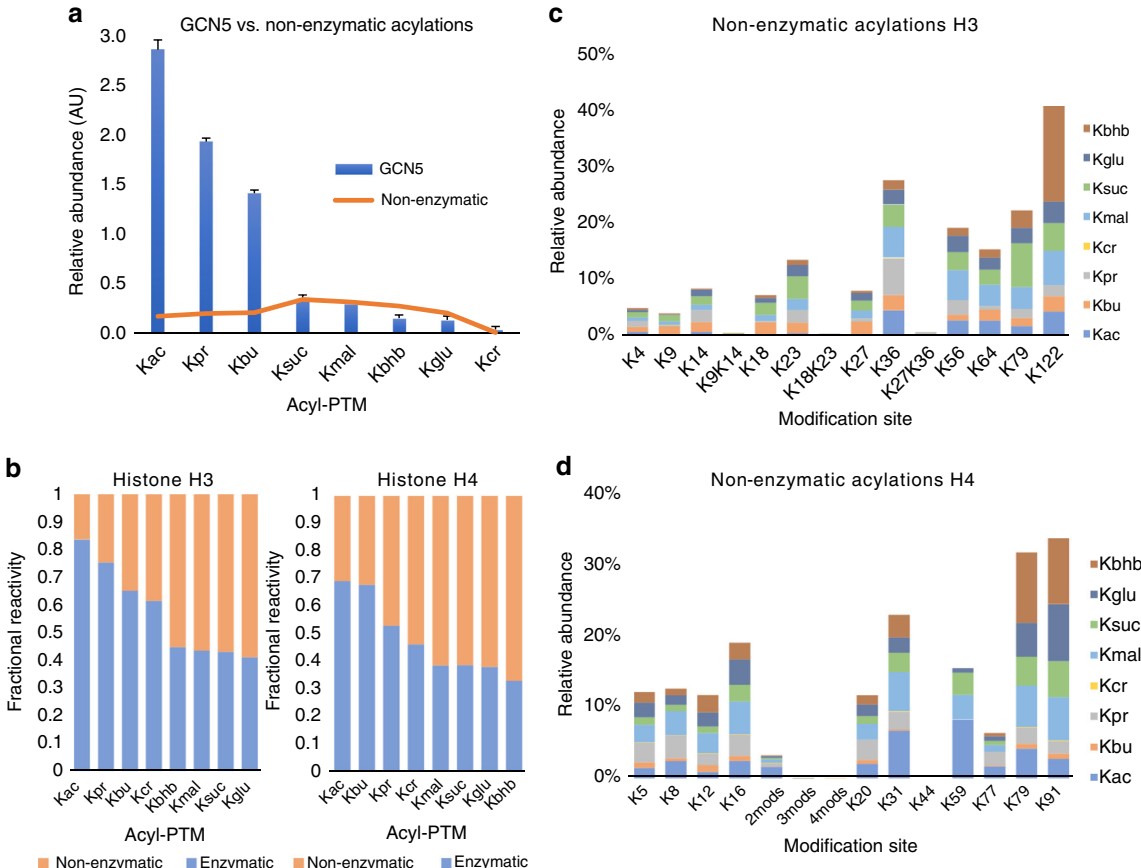

**Fig. 2** Non-enzymatic versus enzymatic acylation of histones in vitro. **a** Comparison of non-enzymatic versus GCN5-catalyzed acylations on histone H3. The y axis (arbitrary units) represents the sum of the relative abundances of all enzymatically and non-enzymatically acylated peptides from histones H3. We can observe that the contributions for Ksuc, Kmal, Kbhb, Kglu and Kcr in the experiments with GCN5 were mostly the result of non-enzymatic acylations. **b** Stacked column representation of non-enzymatic reactivity divided by enzymatic reactivity of the eight acyl-CoA donors on histone H3 (left) and histone H4 (right). The fractional reactivity represents the ratio of PTM intensity in presence of all seven enzymes tested versus PTM intensity in absence of enzymes, i.e., 0.5 corresponds to identical intensities with and without enzyme. For instance, crotonylation is an overall low abundance PTM, although the majority detected on histone peptides is the result of an enzymatic catalysis. **c**, **d** Bar plot showing the relative quantitation of non-enzymatically acylated sites on **c** histones H3 and **d** histone H4. All results are shown as the average of three independent experiments and error bars represent the S.D.

before in terms of substrate preference (Fig. 1d). However, individual acylation activities suggest that, while MOF seems to follow the same trend when looking at the sum of all acylated peptides, Tip60 prefers to utilize butyryl-CoA as a cofactor, followed by succinyl-CoA and acetyl-CoA (Supplementary Table 1). Nonetheless, all results reported in Fig. 1 are based on the average contribution of both groups of HATs rather than individual acylation activities. Detailed acylation site specificities for all HATs can be found in Supplementary Table 1. Our data also showed that the N-terminal acetyltransferase NatA can catalyze N-terminal propionylation and butyrylation of histone H4 in vitro (Supplementary Fig. 2).

**Histones are non-enzymatically acylated in vitro.** Non-enzymatic acylation of proteins has been reported to occur through the nucleophilic attack of the unprotonated ε-amino group of lysine residues to the acyl group of acyl-CoAs[28,35]. This mechanism is facilitated by an alkaline pH and high levels of acyl-CoA. In mitochondria, where both conditions are met and evidence of the existence of acetyltransferases is lacking, a large body of evidence suggests that high levels of protein acylation observed in this organelle are the result of non-enzymatic mechanisms[36,37]. To test whether histones can be

non-enzymatically modified in vitro, we incubated histones H3 and H4 with 0.5 mM acyl-CoAs in the absence of acetyltransferases. We found that all acyl-CoAs can chemically acylate histones, as seen in Fig. 2a, showing a comparison between the non-enzymatic acylation profiles of various acyl-CoA donors on histone H3 with acylations mediated by GCN5. We showed that most non-enzymatically catalyzed acylations have site specificities that were different from those enzymatically modified sites (Figs. 1e, 2c, d). The most prevalent sites observed for non-enzymatic acylations on histone H3 were K36, K56, K64, K79 and K122 (Supplementary Fig. 1; Fig. 2c). On H4, the most prevalent sites were K31, K59, K79 and K91 (Supplementary Fig. 1; Fig. 2d). Interestingly, these sites are closer to the C terminus of histones, whereas enzymes showed higher specificity for sites at the N-terminal tails (Fig. 1e). These observations suggest that non-enzymatic acylations may be enhanced by some level of structural and conformational dynamics on histones. In agreement, most succinylated and malonylated sites identified by Xie et al.[38] in vivo have been reported to occur also at the globular domain and C terminus of histones H3 and H4 rather than at the N-terminal tails.

We then estimated which acylations are more likely to occur through enzymatic or non-enzymatic reactivity in vitro. As seen in Fig. 2b, the average contribution for acetylation,

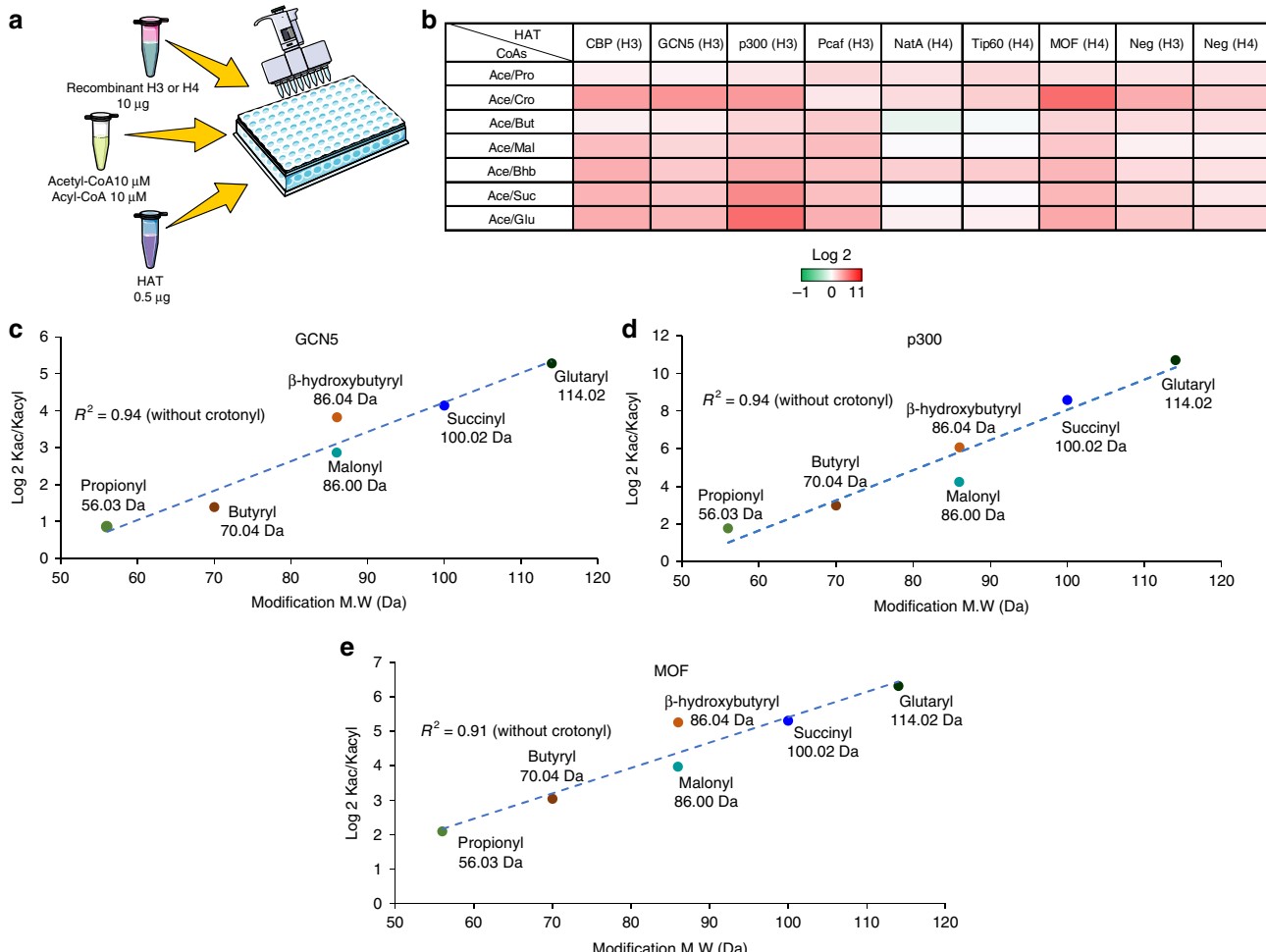

**Fig. 3** In vitro acetylation competition assay. **a** Schematic representation of the in vitro competition acetylation assay. **b** Heat map displaying in vitro acylation specificities of HATs during acyl-CoA competition assays. Different HATs were assayed against histones H3 or H4 as specified in the table headers in the presence of equimolar concentrations of acetyl-CoA and a competing acyl-CoA donor. Negative controls with no enzyme are also shown. **c** Correlation between the molecular weight and the acylation preference for different acyl donors displayed for the HATs GCN5 on histone H3, **d** p300 on histone H3 and **e** MOF on histone H4. For each modification, molecular weight is indicated. Results show that the preference for acetyl-CoA over the other acyl donor tightly correlates with the molecular weight of the acyl donor. Crotonylation was not included in the correlations, as its molecular weight did not correlate well with the preference of the enzymes over acetyl-CoA. All graphs are shown as the average of log2 ratios between the relative abundances of all acetylated peptides and the relative abundances of the corresponding competing acylated peptide

propionylation and butyrylation marks in the presence of all HATs was more abundant in in vitro experiments, whereas acidic acyl modifications (malonylation, succinylation and glutarylation) and β-hydroxybutyrylation occurred to a greater extent through non-enzymatic mechanisms in both histones H3 and H4. Again, we observed a trend in which the ratio of enzymatic/chemical reactivity of acyl groups is inversely correlated with the size of the side chain, supporting the idea that most known HATs catalyze larger acylations less efficiently[32,33]. When looking individually at non-enzymatic acylations in histones, crotonyl-CoA showed the lowest acylation levels through chemical catalysis (Fig. 2c, d). As expected, crotonyl-CoA presents a lower chemical reactivity towards lysine nucleophiles due to the resonance properties of the beta unsaturated C-C bond[39].

The correlation between sites prone to chemical acylation (this study) and sites identified in other in vivo studies suggests that histone acyl modifications in cells could be the result of both enzymatic and non-enzymatic mechanisms after direct exposure to intrinsically reactive acyl-CoA metabolites. This is supported by the fact that enzymes with distinctive acyltransferase activities have not been identified in any cellular compartment[40]. However,

it is important to mention that concentrations of acyl-CoAs used in this experiment were far above the known physiological concentrations of CoA derivatives in whole cells[41], so the extent of chemical acylation observed in this study is likely an overestimation. This higher concentration was required to ensure proper sensitivity to the in vitro assay. Even though it has been previously demostrated that protein lysine acylation can occur at physiological acyl-CoA concentrations in vitro[42], our study does not represent a suitable extrapolation for the reactivity of these intermediates in cells. As such, our data cannot rule out the possibility of the existence of enzymes that play a major role in catalyzing these marks in vivo as compared to non-enzymatic reactions.

**HATs prefer acetyl-CoA for acylation of histones**. Previously, it has been demonstrated that some HATs are able to catalyze the transfer of propionyl and butyryl groups in vitro with similar specificites but different efficiencies than acetyl groups[12,31]. These observations lead to the following question: what determines the acyl group to be transfered if HATs are indeed mediating

various acylations in vivo? It has been shown that fibroblasts of patients with inherited metabolic disorders have high levels of propionyl-, malonyl- and butyryl-CoA and high levels of the corresponding lysine acylations[43,44]. These disorders include deficiencies in propionyl-CoA carboxylase (PCC), malonyl-CoA

decarboxylase (MCD) and short-chain acyl-CoA dehydrogenase (SCAD), respectively. However, the mechanism by which this increase in protein acylation is mediated has not been explored. We hypothesized that under such conditions, other acyl-CoAs could rival the levels of acetyl-CoA and induce HATs to use non-

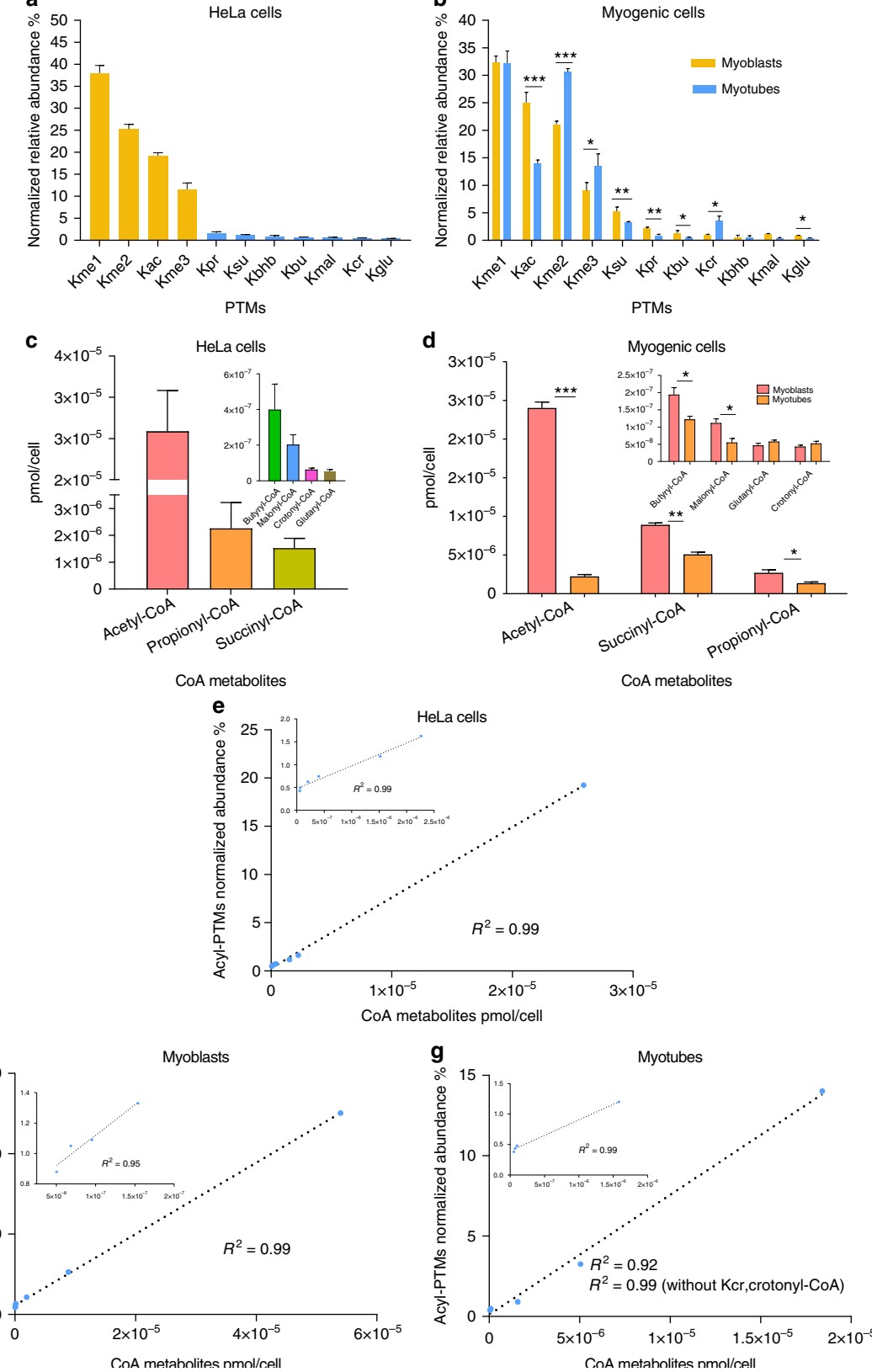

native cofactors. To test this, we performed in vitro HAT competition assays in the presence of equimolar concentrations of acetyl-CoA and other acyl-CoAs (Fig. 3a). As shown in Fig. 3b, most HATs preferred to utilize acetyl-CoA than any other acyl-CoA donor. It is important to mention that at 10 μM of acyl-CoAs, we observed non-enzymatic acylation of histones H3 and H4, as shown in Fig. 3b. Consistent with previous in vitro experiments, for most HATs, the preference for the competing cofactor, if any, largely depended on the size of the acyl donor side chain. As shown in Fig. 3c–e, for the enzymes GCN5, p300 and MOF we observed an inverse correlation between the HAT preference for the competing cofactor and the increasing molecular weight of the acyl donor side chain, with the exception of crotonyl-CoA. Relative abundances for all peptides in competition assays are shown in Supplementary Table 3. Altogether, our data suggest that even in the highly unlikely chance that any other acyl-CoA accumulated to the extent that its concentration rivaled that of acetyl-CoA, HATs would still mostly utilize acetyl-CoA. We thus shifted our focus to investigate how abundant these acyl marks are in vivo, and whether their abundance can be justified by the abundance of acyl-CoA intermediates.

**Relative abundances of histone acyl-PTMs in mammalian cells.** Owing to the low abundance of lysine acyl marks, current MS-based approaches involve the use of antibody-based enrichments to increase the sensitivity of the MS analysis for identification and quantification. Although these approaches are helpful for estimating the relative changes of modifications across multiple conditions, they cannot provide direct information on relative abundances, as the peptide with the modification and the unmodified peptide end up in different sample pools. In addition, evaluation of the specificity of several commercially available pan anti-acyl-PTM antibodies by dot blot analysis revealed significant cross-reactivity among differentially acylated peptides, complicating their further application for immunoenrichment of histone acyl-PTMs (Supplementary Fig. 3).

Thus, to accurately detect and quantify histone acyl-PTMs in vivo using label-free approaches, we used the retention time and mass shift information from the in vitro experiments to optimize the MS acquisition and in-house quantification software[45]. Here, we refer to our quantitative values as "relative abundances", as we are aware that differences in the ionization efficiencies of modified peptides or biases in trypsin digestion in the presence of certain modifications can affect the assessment of accurate PTM stoichiometry, which was observed for the differentially acylated peptide H3 aa 18–26 (KQLATKAAR) (Supplementary Fig. 4).

Analysis of acid-extracted histones from wild-type HeLa and myogenic cells showed that all acyl-PTMs combined, excluding acetylation, were found at relative abundances between 6 and –15% of all detectable modified peptides of canonical histones H3 and H4 (Supplementary Fig. 5). Individually, most acyl-PTMs showed low relative abundances ranging from 1 to –5% (Figs. 4a, b), as opposed to acetylation with global relative abundances

between 15 and –30% (Fig. 4a,b; Supplementary Fig. 5). Using our workflow, we measured the relative abundances at individual sites. Acetylation was found at high relative abundances at positions H3K18 and H3K23 ranging from 17 to –35%, and between 15–30% at H4K12 and H4K16 (Supplementary Table 2), which is in accordance with previous findings[46]. Interestingly, similar relative abundances were observed at H3K14 (3–8%) for acetylation and propionylation marks (Supplementary Table 2). Overall, most non-acetyl acyl marks were found at levels below 2%, mainly at the N-terminal domains of histones H3 and H4. Surprisingly, some acidic acyl marks were found at the globular domains and C terminus, showing abundances as high as 10% (Supplementary Table 2), which coincide with the sites that were more susceptible to non-enzymatic acylation in our in vitro experiments (Fig. 2c, d). Detailed site specificity for all acyl marks is shown in Supplementary Table 2. Importantly, we did detect low levels of hydroxybutyrylation; however, our MS acquisition method cannot discriminate between possible isoforms of this mark, including bhb (β-hydroxybutyryl or 3hb), 2-hydroxybutyryl (2hb), 3-hydroxyisobutyryl (bhib), 2 hydroxyisobutyryl (2hib) or 4-hydroxybutyryl (4hb)[10,14]. Likewise, peptides bearing Kbu (butyryl) marks may also represent isobutyryl marks.

Our analysis of myoblasts showed dynamic changes in global histone acylations upon their fusion to form multinucleated myotube cells. We observed that the global levels of lysine acetylation, propionylation, butyrylation, malonylation, succinylation and glutarylation were significantly decreased upon myogenic differentiation, whereas the levels of lysine crotonylation were increased (Fig. 4b). The role of differential histone acylation in cellular differentiation is poorly understood; however, various lines of evidence suggest that nutrition and metabolism play a key role in the differentiation of cells[47,48]. For example, a previous study investigating the role of carbon metabolism in the differentiation of myogenic cells demonstrated that siRNA knock down of ATP citrate lyase (ACLY) induces differentiation of mouse myoblasts[49]. The same study also showed that the levels of histone acetylation in shACLY-treated cells were reduced, hypothesizing that the deposition of acetyl-CoA, and in turn histone acetylation levels, play an important role in the differentiation of myoblasts. Because there is little evidence of how newly identified histones acylations may be implicated in the differentiation of cells and/or epigenetic regulations, we next sought to investigate whether the cellular concentrations of other acyl-CoA metabolites have a direct relationship with histone acylation levels.

**Acyl-CoA donors dictate the levels of histone acylation.** Various studies in mammalian cells have shown that chromatin modifications are sensitive to changes in intracellular concentrations of metabolic intermediates, linking cell metabolism to epigenetic changes[6,50]. However, the mechanisms and enzymes mediating these processes have not been fully explored. So far, it has been demonstrated that changes in the levels of acetyl-CoA can influence global histone acetylation levels[51]. These findings led us

**Fig. 4** Overview of the relative abundances of endogenous acyl-PTMs and intracellular metabolite concentrations. Bar plots showing the relative abundances of several lysine PTMs in **a** HeLa cells, **b** proliferative myogenic cells (myoblasts) and differentiated myogenic cells (myotubes). PTMs are shown as percentages representing the normalized relative abundances of all detectable peptides of canonical histones H3 and H4. **c, d** Bar plots showing the concentrations of acyl-CoA metabolites in **c** HeLa cells and **d** myogenic cells normalized to cell number. Metabolic concentrations of butyryl-CoA may also represent concentrations for isobutyryl-CoA. **e–g** Global histone acylation levels correlation with intracellular concentrations of acyl-CoA metabolic intermediates in **e** HeLa cells, **f** myoblasts and **g** myotubes. Correlations are calculated between the normalized net abundances of all detectable acylated peptides in H3 and H4 and the global concentrations CoA metabolites in pmol per cell. Insets show zoom in graphs with the linear correlation of the data without the values for acetyl-CoA and Kac. All results are shown as the average of three biological replicates and error bars represent the S.D. Summary of *p*-values is as follows; *$p \le 0.05$, **$p \le 0.01$, and ***$p \le 0.001$. *p*-values were generated by unpaired Student's *t*-test

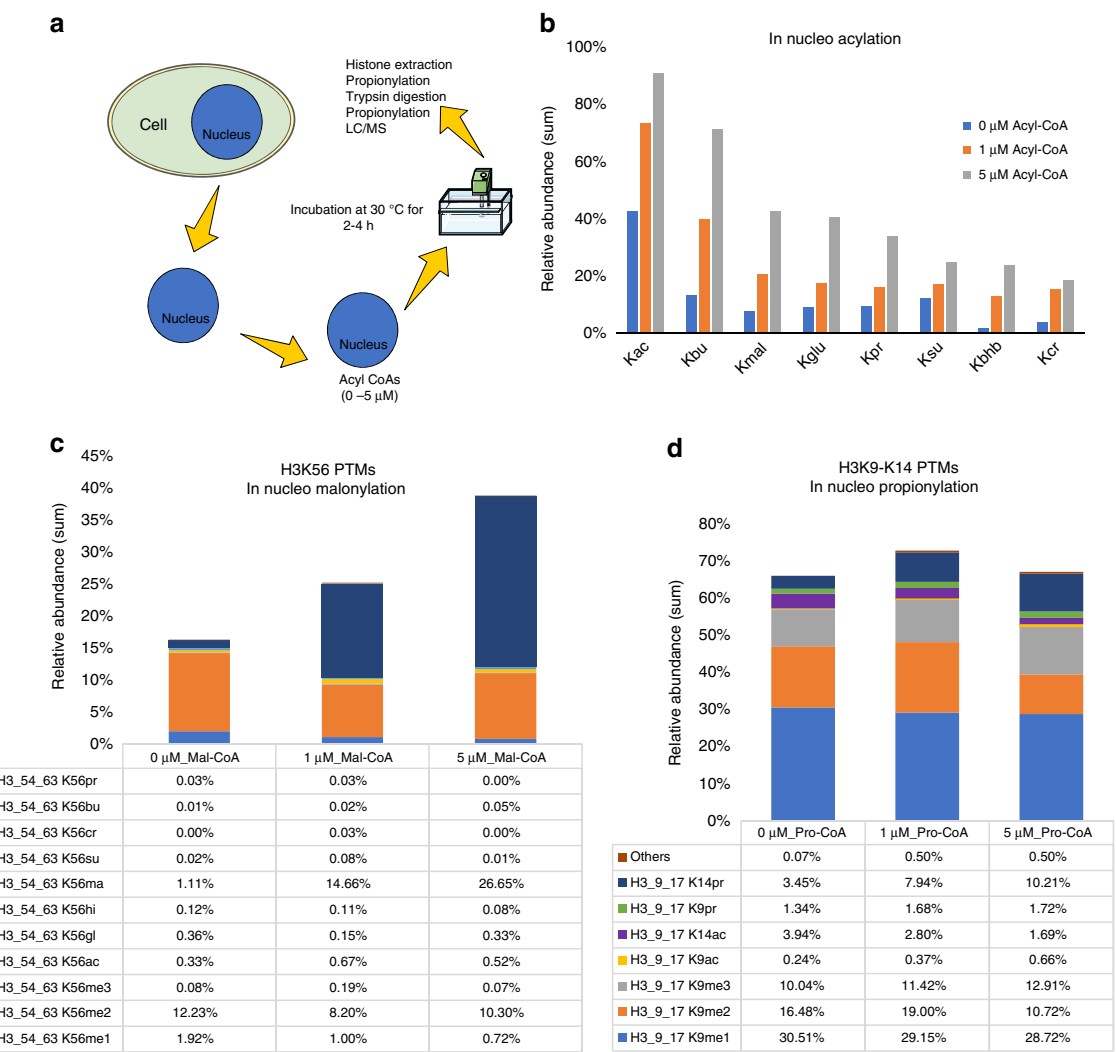

**Fig. 5** Analysis of in nucleo acylation. **a** Schematic representation of in nucleo acylation assay. **b** Bar plots showing the dose-dependent acylation of histones upon treatment with 0, 1 and 5 μM of acetyl-, butyryl-, malonyl-, glutaryl-, propionyl-, succinyl-, β-hydroxybutyryl and crotonyl-CoA, respectively. Values represent the sum of the relative abundances of all acylated peptides from histones H3 and H4. **c** Example of in nucleo acylation resulting in an increase in malonylation of H3K56 after treatment with increasing concentrations of malonyl-CoA. **d** Example of in nucleo acylation showing that induced propionylation by increasing the concentration of propionyl-CoA resulted into a reduced relative abundance of acetylation on the site H3K14

to consider whether the levels of other histone acylations may also be influenced by the intracellular concentrations of their respective acyl-CoA donors. Using a stable isotope dilution MS approach[52], we accurately measured the concentrations of seven acyl-CoA metabolic intermediates in HeLa (Fig. 4c) and myogenic cells (Fig. 4d). Our metabolomics analysis revealed that acetyl-, propionyl- and succinyl-CoA were the most abundant CoA thioesters in our cell models, with concentrations around 12, 1 and 0.5 μM, respectively, when normalized to cell volume of HeLa cells (Supplementary Fig. 6). In myogenic cells, when normalizing the data to cell number, we observed that the intracellular concentrations of metabolites also appear to undergo regulation from myoblast to myotube differentiation. Our data showed that the levels of acetyl-, succinyl-, propionyl-, butyryl and malonyl-CoA were significantly decreased upon differentiation (Fig. 4d), whereas the levels of glutaryl- and crotonyl-CoA did not show significant changes.

When looking at the levels of acyl lysine PTMs in HeLa cells, our data showed a strong positive correlation with the metabolic levels of acyl donors (Fig. 4e). A similar observation could be made for myoblasts (Fig. 4f). Even though we observed a

measurable increase in the levels of crotonyl-CoA, this did not correlate well with the increase in lysine crotonylation observed in myotubes (Fig. 4g). Acyl-CoA intermediates are derived from various metabolic pathways including the TCA cycle, fatty acid synthesis, β-oxidation and amino acid metabolism. Although it remains poorly understood which metabolic pathway leading to the production of different acyl-CoAs might serve as a substrate for the acylation of nuclear histones, the global concentrations of metabolites determined in this study are consistent with the abundance of lysine acylations. Collectively, our data demonstrate a clear quantitative link between metabolism and differential histone acylations.

**Acylation of histones in nucleo.** Our in vivo and metabolomics studies strongly implicated metabolism in histone acylations. As the concentrations of intracellular metabolites are known to change in response to diet or physiological conditions, we next turned to the question of whether alterations in the levels of metabolites could affect the corresponding levels of histone acylations. To further explore this idea, we turned to an in nucleo

system, where purified nuclei can be treated with artificial levels of metabolites. We isolated nuclei from HeLa cells under hypotonic conditions, treated with varying concentrations of eight different acyl-CoA donors and performed histone extraction, digestion and derivatization following standard procedures (Fig. 5a).

MS analysis revealed that histone acylations can be induced in a concentration-dependent manner. Specifically, by adding 1 or 5 μM of acyl-CoAs, we induced an increase of the respective acylation on histone peptides (Fig. 5b). Since the in nucleo experiment preserves the natural state of nuclear processes, it can be used to observe histone acylations in native chromatin. Our in vitro experiment demonstrated that acylations can occur by both enzymatic and non-enzymatic mechanisms, but such simplified assay cannot accurately represent the balances of a nuclear environment. We compared the two assays by performing an in nucleo–in vitro Spearman's rank-order correlation analysis by using corrected in vitro enzymatic data (subtracting the non-enzymatic contribution). We observed a good correlation for some residues, including H3K9acyl and H3K18acyl sites (Supplementary Fig. 7) that were highly acylated only in presence of enzymes in vitro. This suggests that specific sites are likely more accessible to enzymatic activity than others, and that this reactivity is also a function of the acyl-CoA utilized. However, a generalized conclusion cannot be drawn, as the in nucleo assay cannot discriminate enzymatic catalysis from chemical reactions, and physiological acylation turnover (equilibrium deposition/removal).

Additionally, our in nucleo results showed that histones accommodate acylation in two ways; by simply increasing the modified state or by removing pre-existing modifications to maintain the same level of total modified form. For example, upon treatment with malonyl-CoA, levels of H3K56mal increased with almost no changes in the other modifications on that peptide (Fig. 5c). On the other hand, after treatment with increasing concentrations of propionyl-CoA, the levels of H3K14pr increased, whereas the levels of H3K14ac showed a measurable decrease (Fig. 5d). Detailed in nucleo acylation relative abundances can be found in Supplementary Table 4. Taken together, our in nucleo studies demonstrated that modifications in chromatin are sensitive to changes in the concentrations of cellular metabolites, consistent with previous observations connecting the metabolic state of the cell with chromatin regulation[7,53].

## Discussion

A comprehensive screen of the major families of histone acetyltransferases (HATs) confirmed that most enzymes can catalyze the acylation of histones utilizing acetyl, propionyl and butyryl-CoA cofactors with similar efficiencies (Fig. 1), as previously reported[54]. However, they were less efficient catalyzing the acylation of histones with charged, branched or planar acyl-CoA cofactors. Even though these acyl donors are structurally similar to acetyl-CoA, the universal cofactor of HATs, our data showed that the ability of enzymes to utilize other cofactors largely depended on the size of the acyl group, which is in close agreement with recent data demonstrating the structural incompatibility of the active sites of p300 and GCN5 with long-chain acyl donors[32,33]. This observation was further confirmed by in vitro HAT assays performed at equal concentrations of acetyl-CoA and a competing cofactor (Fig. 3). The data showed a similar trend in which the preference for a cofactor different from acetyl-CoA was inversely proportional to the molecular weight of the competing donor, except for crotonyl-CoA, which, unlike the other acyl groups, possesses an unsaturated moiety that seems to render its use by most HATs unfavorable.

To further explore the possible mechanisms underlying the establishment of acyl marks, we performed in vitro acylation assays in the absence of HATs (Fig. 2). We found that histones can be acylated by the chemical reactivity of all acyl-CoA metabolites evaluated in this study. While most HATs showed strong preference for the acylation of residues at the N-terminal domains, i.e., residues K9–K36 in H3 and K5–K16 in H4, non-enzymatic acylation sites were more prevalent closer to the C terminus of histones, i.e., residues K56–K122 in histone H3 and K59–K91 in histone H4. Acidic acyl-PTMs including malonylation, succinylation and glutarylation were among those most easily catalyzed in the absence of enzymes (Fig. 2b). These marks are different from acetyl marks as they add bulkier groups to lysine residues and they carry a negative charge under physiological conditions. As such, it has been suggested that these acidic acyl marks could disrupt the interactions between histones and DNA, resulting in a more profound effect in chromatin unfolding than lysine acetylation[55]. Emerging hypotheses have suggested a model where non-enzymatic chemical reactions are a significant contributor to the landscape of lysine acylations in nuclear histones[56]. They also suggest that sirtuin enzymes showing specificity for the removal of acyl marks may represent a constitutive programming to suppress potential damaging effects caused by the presence of these PTMs[56,57]. Interestingly, our study showed that those succinylated and malonylated sites highly susceptible to non-enzymatic acylation in vitro were among the sites reported previously in in vivo studies[38]. While more studies are required, our data suggest that histone lysine residues are prone to be modified by several free acyl-CoAs with and without enzymatic assistance.

The accurate quantification and elucidation of potential functional roles of acyl marks have been hampered by their low abundance. To provide an accurate estimate of the levels of acyl marks, we employed a label-free approach using DIA-MS. By analyzing in the same mixture modified and unmodified forms of histone peptides, we could report the relative levels of acyl-PTMs expressed as a percentage of the total histone. Analysis of human cervical cancer cells (HeLa) and human myogenic cells revealed that acyl marks together represent around 6–15% of all detected modifications on histones H3 and H4 (Fig. 4a,b; Supplementary Fig. 5). Our myoblast/myotube comparison strongly indicates that the differentiation of pluripotent cells is marked by a decrease in global levels of histone acetylation. This was not surprising, as this mechanism has been shown to be driven by a decrease in acetyl-CoA production mediated through the inhibition of glycolysis[51,58]. In agreement with these findings, our data showed a significant decrease of the bulk levels of histone acetylation once myoblasts fused to form multinucleated myotubes. Intriguingly, while the levels of most histone acylations decreased upon differentiation, myotube cells showed an increase in the global levels of lysine crotonylation (Fig. 4b). The understanding of how these chromatin modifications could be involved in driving myoblast differentiation is beyond the scope of this study and the subject of studies to come. This is a comprehensive report providing quantitative information on the levels of a broad number of histone acyl marks in HeLa and human myogenic cells, thus representing an important resource for future work aiming to understand cellular function and the dynamics of acyl-PTMs in the complex mammalian epigenetic mechanisms.

In general, we observed a strong correlation between histone acylations and their corresponding metabolic substrates. One exception was crotonylation, as the observed increase in lysine crotonylation in myotubes was not accompanied by a statistically significant increase in the levels of crotonyl-CoA (Fig. 4b,d,g). This specific experiment cannot prove whether crotonylation in myotubes is regulated by enzymatic mechanisms

that are not strictly regulated by crotonyl-CoA levels. However, we performed an intermediate experiment between in vivo and in vitro, namely in nucleo, to test whether there are corresponding changes in the levels of histone acylations upon manipulation of the concentrations of metabolites. This experiment showed that the bulk levels of lysine acylations can be induced in a dose-dependent manner, resulting in either a net increase or a dynamic exchange of modifications in response to increasing concentrations of acyl-CoA metabolites (Fig. 5). Despite the evidence that the concentrations of metabolites can regulate the global levels of lysine acylations, it is premature to pinpoint how this metabolic regulation is involved in complex processes including gene expression, cell differentiation and apoptosis, or in diseases such as cancer that are characterized by altered metabolic states. Such a scenario is further complicated considering that metabolite precursors of histone modifications exist in different pools derived from various biological pathways that are regulated in response to cellular physiological conditions. Continued work in this area will help elucidate remaining questions surrounding the role of acyl-PTMs, such as the following: (i) do these PTMs compete with lysine acetylation, or (ii) do they work in concert to regulate epigenetic mechanisms? Various studies have shown that the availability of acetyl-CoA can regulate the abundance of lysine acetylation. Our conclusion is that this study provides compelling evidence that this metabolic regulation is not restricted to acetylation, but also extends to the regulation of newly identified acyl marks in chromatin.

## Methods

**Protein expression and purification.** Human p300 and human CBP sequences containing an N-terminal His tag, and C-terminal Strep2 and FLAG tags, were synthesized and cloned by Genewiz (Cambridge, MA) into the pVL1393 vector for baculovirus expression. The plasmid was transfected into Sf9 cells utilizing the BD (Franklin Lake, NJ) BaculoGold transfection system. Then, p300 was expressed in Sf9 cells and purified using a GE Healthcare (Piscataway, NJ) HiTrap column. The protein identity and purity were confirmed through protein staining with Coomassie dye[59].

The NatA (Naa10p/Naa15p) protein complex from *Schizosaccharomyces pombe* was prepared essentially as previously described[60]. Briefly, protein expression vectors encoding 6XHis-tagged Naa15p and residues 1–156 of Naa10p were overexpressed in Rosetta (DE3) pLysS *E. coli* cells, and purified to homogeneity using a combination of Ni-resin affinity, 6XHis-TEV protease treatment to remove the 6XHis-tag, Ni-resin affinity to remove the 6XHis-TEV protease, HiTrap SP ion exchange and Superdex 200 gel filtration. The protein complex was concentrated to ~ 10 mg per mL in a buffer containing 25 mM HEPES (pH 7.0), 200 mM NaCl and 1 mM TCEP and stored frozen at −70 °C until use.

The HAT domain of human hMOF was prepared previously essentially as described[61]. Briefly, a protein expression vector encoding 6xHis-tagged hMOF HAT domain (residues 174–449) was overexpressed in *Escherichia coli* cells and the protein purified to homogeneity using a combination of Ni-resin affinity and HiLoad Superdex 75 gel filtration. The protein was concentrated to ~ 20 mg per mL in a buffer containing 20 mM HEPES (pH 7.5), 0.5 M NaCl, and stored as described for the NatA complex.

The HAT domain of *S. cerevisiae* Gcn5 (residues 99–262) was overexpressed in bacteria from a PRSETA/yGCN5 vector and purified similarly as previously described[62]. The plasmid was transformed into *E. coli* strain BL21 (DE3) and overexpressed by induction with 0.5 mM isopropyl-β-D-thiogalactopyranoside (IPTG) and grown at 17 °C overnight. The cells were collected by centrifugation at 4000 r.p.m. at 4 °C and lysed in 50 mM potassium phosphate pH 7.5, 0.500 M NaCl, 5% glycerol, 1 mM dithiothreitol (DTT), 0.1 mM phenylmethylsulfonyl fluoride (PMSF), 100 ug per mL DNase I, and 100 μg per mL lysozyme. The supernatant liquid was passed over Ni-NTA resin, washed with 10 column volumes of lysis buffer with 5 mM imidazole but without PMSF, DNase, or lysozyme. The protein was then transferred to 6–8 kDa MWCO dialysis tubing (Spectrum Labs) and dialyzed overnight into 50 mM potassium phosphate pH 7.5, 20 mM NaCl, 5% glycerol, and 1 mM DTT buffer. This was then followed by additional purification through SP Sepharose ion exchange and Superdex 75 gel filtration. Purified protein was concentrated to ~ 20 mg per mL in buffer containing 50 mM potassium phosphate, pH 7.5, 500 mM NaCl, 5% glycerol and 1 mM DTT, and flash frozen and stored at −70 °C.

The DNA sequence encoding residues 443–658 (including an N-terminal Met residue) of human PCAF was amplified by PCR and subcloned into the pET28-A vector (Invitrogen) for overexpression similarly as previously described[63]. The plasmid was transformed into *E. coli* strain BL21 (DE3) and overexpressed by

induction with 0.5 mM IPTG and grown at 17 °C overnight. The cells were collected by centrifugation at 4000 r.p.m. at 4 °C and lysed in 25 mM HEPES pH 7.5, 0.150 M NaCl, 1 mM DTT, 0.1 mM phenylmethylsulfonyl fluoride, 100 ug per mL DNase I, and 100 ug per mL lysozyme. PCAF was purified from the lysate through Ni-NTA affinity as described above with 5 mM imidazole in 25 mM HEPES pH 7.5, 0.150 M NaCl, 1 mM DTT and eluted with 200 mM imidazole in the same. The protein was then transferred to 6–8 kDa MWCO dialysis tubing (Spectrum Labs) and dialyzed overnight into 20 mM sodium citrate pH 6.0, 0.150 M NaCl, and 1 mM DTT buffer. This was then followed by additional purification through SP Sepharose ion exchange and Superdex 75 (GE) gel filtration. The protein was concentrated to ~ 30 mg per mL, flash frozen and stored at −70 °C in a buffer containing 20 mM Na-citrate pH 6.0, 150 mM NaCl, 1 mM DTT.

Recombinant histones H3 and H4 were expressed in Rosetta BL21 [DE3] pLysS cells and purified as monomers using standard procedures[64,65].

**In vitro histone acylation assay.** Based on substrate specificity, histone H3 was assayed with HATs p300, CBP, PCAF and Gcn5, and histone H4 with HATs MOF, NatA and Tip60. In vitro enzymatic assays were carried by incubating 0.5 μg of each HAT with 10 μg of recombinant histones H3 or H4 in the presence of 0.5 mM of short-chain acyl-CoAs (acetyl-, crotonyl-, malonyl-, succinyl-, propionyl-, butyryl-, glutaryl- and β-hydroxybutyryl CoA;—Sigma-Aldrich) in 1X HAT buffer (25 mM Tris-HCl pH = 8, 25 mM KCl, 1 mM DTT, 0.1 mM AEBSF and 5 mM sodium butyrate) for 60 min at 30 °C; the final volume was 50 μL. For competition assays, reactions were carried out in the presence of 10 μM of acetyl-CoA and 10 μM of other acyl-CoAs. Background control reactions were performed in the absence of HATs. Reactions were stopped by freezing and samples were dried in a SpeedVac and resuspended in 20 μL of 100 mM ammonium bicarbonate (pH 8.0). Histones were derivatized with propionic anhydride (or d$_{10}$-propionic anhydride when analyzing lysine propionylation). For this procedure, fresh propionylation reagent was prepared by mixing propionic anhydride with acetonitrile in the ratio 1:3 (v/v). Propionylation reagent was added to each sample in 1:4 (v/v). Ammonium bicarbonate was quickly added to the solution to re-establish pH 8.0. Samples were dried down to 10–20 μL in a SpeedVac, reconstituted with 20 μL of ammonium bicarbonate and the propionylation procedure was repeated one more time. Samples were then digested with trypsin at a 1:10 ratio (wt/wt) for 6 h at 37 °C. Samples were desalted by C18 stage-tip. For this procedure, a small piece of a C18 solid phase extraction disk was deposited into a pipette tip to create a stage-tip. The C18 resin was flushed with 100% acetonitrile by slow centrifugation using a centrifuge adaptor to hold the stage-tips in place in a 1.5 ml Eppendorf tube. The resin was then equilibrated by flushing 80 μl of 0.1% TFA. Samples were acidified to pH 4.0 or lower with acetic acid and loaded onto the disk by slow centrifugation. Samples were then washed by flushing 70–80 μl of 0.1% TFA and eluted into a clean Eppendorf tube by flushing 70 μL of 75% acetonitrile and 0.5% acetic acid by slow centrifugation. Samples were dried in a SpeedVac and resuspended at 0.5 μg per μL in 0.1 M acetic acid for nano LC-MS/MS analysis[66].

**Cell culture and histone extraction.** HeLa S3 mammalian cells were cultured at 37 °C and 5% CO$_2$ in spinner flasks in Joklik's modified Eagle's medium supplemented with 10% (v/v) newborn calf serum (HyClone), penicillin-streptomycin (1:100), and 1% (v/v) GlutaMAX (Invitrogen). Cells were harvested, washed with PBS, and stored at −80 °C. Human myogenic cell line, LHCN-M2 (a kind gift from Dr. Woodring Wright, UT Southwestern Medical Center at Dallas, Dallas, TX, USA) was cultured in proliferation medium described elsewhere[67]. In brief, confluent cells were differentiated in 2% horse serum. Matched cultures for proliferation and differentiation were set up, the myoblasts harvested at 80% confluence and differentiation cultures harvested on day 5–7 when the myotubes are formed fully. Briefly, 0.05% trypsin-EDTA was used to detach cells after washing with sterile PBS. The cells were collected by centrifugation, washed again in PBS, snap-frozen in liquid nitrogen, and stored at −80 °C until further analysis. Three biological replicates were used. Histone extraction and digestion were carried out according to standard procedures[66]. Briefly, nuclei were isolated by suspension of cells in 10X volume of nuclear isolation buffer (15 mM Tris-HCl pH = 7.5, 60 mM KCl, 15 mM NaCl, 5 mM MgCl$_2$, 1 mM CaCl$_2$ and 250 mM sucrose, 0.2% NP-40) with 1 mM DTT, 5 nM microcystin, 0.5 mM ASBSF and 10 mM sodium butyrate at 4 °C. Nuclei were pelleted by centrifugation at 1000× g for 5 min at 4 °C and washed twice with nuclear isolation buffer in the absence of NP-40. To the pelleted nuclei, 0.4 N H$_2$SO$_4$ was added to a final ratio of 5:1 and incubated for 2 h with shaking at 4 °C. Samples were centrifuged at 3400× g for 10 min at 4 °C and the supernatants were collected and incubated on ice with ¼ volume of 100% TCA for 1 h. Precipitated histones were collected by centrifugation at 3400× g for 10 min at 4 °C and pellets were rinsed once with ice-cold acetone containing 0.1% HCl and once with ice-cold acetone. Protein concentration was determined using a Bradford assay. For in vivo analyses of histone acyl marks, 50 μg of histones were resuspended in 20 μL of 50 mM ammonium bicarbonate and subjected to derivatization with propionic anhydride (or d-10 propionic anhydride when identifying propionyl and butyryl histone marks), digested with trypsin for 6 h at 37 °C and desalted with C18 stage-tips as described above.

**In nucleo acylation assay**. HeLa S3 mammalian cells were harvested and washed twice in ice-cold PBS. Cells were incubated on ice with hypotonic lysis buffer containing 10 mM Hepes-NaOH pH = 7.9, 10 mM KCl, 1.5 mM $MgCl_2$, 0.5 mM DTT, 0.1% (v/v) NP-40, and 1X Halt protease and phosphate inhibitors for 10 min with intermittent agitation. Nuclei were collected by centrifugation at 600 *g* for 10 min at 4 °C, and washed once with a buffer containing 20 mM Hepes-NaOH pH = 7.9, 50 mM KCl, 1.5 mM $MgCl_2$, 0.5 mM DTT, 0.2 mM EDTA, 20% (v/v) glycerol and 1 X Halt protease and phosphate inhibitors. Nuclei were aliquoted at ≈ $5 \times 10^6$ cells per mL and used immediately by resuspending in 50 μL acylation reaction buffer (50 mM Tris-HCl pH = 8.0, 50 mM NaCl, 1 mM EDTA, 0.5 mM DTT, 10% (w/v) sucrose, 1 X Halt protease and phosphate inhibitors), containing 0, 1 or 5 μM of short-chain acyl-CoAs (acetyl-, crotonyl-, malonyl-, succinyl-, propionyl-, butyryl-, glutaryl- and β-hydroxybutyryl CoA). Reactions were incubated at 30 °C for 3 h with gentle agitation. Nuclei were washed with acylation reaction buffer and the reactions were terminated by resuspending nuclei in 0.4 N $H_2SO_4$ for histone extraction. Extracted histones were derivatized with d10-propionic anhydride and digested as described above. Samples were dried in a SpeedVac and resuspended at 0.5 μg per μL in 0.1 M acetic acid for nano LC-MS/MS analysis.

**Dot blot analysis**. 2 μg of full-length acylated histones H3 or H4 from in vitro reactions were spotted onto a nitrocellulose membrane. The membrane was blocked with either 5% BSA or 5% nonfat milk and incubated with the primary antibodies at 1:1,000 dilution (pan anti-crotonyl-lysine: PTM-501, pan anti-propionyl-lysine: PTM-201, pan anti-malonyl-lysine: PTM-901, pan anti-succinyl-lysine: PTM-401, pan anti-butyryl-lysine: PTM-301, pan anti-glytaryl-lysine: PTM-1151, all purchased from PTM-Biolabs, Hangzhou, China) according to the manufacturers' instructions overnight at 4 °C. The membrane was washed with TBS-T three times for 10 min each, incubated with secondary antibody (Goat anti-Rabbit IgG Fc, Pierce 31463) at a 1:10,000 dilution for 60 min at room temperature and then probed with ECL Western Blot Substrate (Pierce).

**Nano LC-MS/MS analysis**. A total of 2.5 μg of peptides were injected into a 75 μm i.d × 17 cm Reprosil-Pur C$_{18}$-AQ (3 μm; Dr. Maisch GmbH, Germany) nanocolumn (packed in-house) using an EASY-nLC nano HPLC (Thermo Scientific, Odense, Denmark). The mobile phases consisted of water with 0.1% (v/v) formic acid (A) and acetonitrile with 0.1% (v/v) formic acid (B). For analysis of in vitro samples, peptides were eluted using a gradient of 0–30% B for 20 min followed by 30–98% B for 5 min and maintained over 10 minutes at 300 nL per min. For nucleo and in vivo samples, the gradient consisted of 0–26% B over 45 min followed by 26–98% B over 5 min and maintained for 10 min at 300 nL per min. The nano HPLC was coupled to a LTQ Orbitrap Elite or an Orbitrap Fusion mass spectrometer (Thermo Scientific, San Jose, California). Spray voltage was set at 2. 4 kV and capillary temperature was set at 275 °C. For DDA, the mass spectrometer was set to perform a full MS scan (290–1,400 *m/z*) in the Orbitrap with a resolution of 60,000 (at 400 *m/z*), followed by a series of targeted MS/MS scans of each modified H3 and H4 peptide, followed by MS/MS scans of the top four most intense abundant ions from the first scan. All MS/MS scans were performed in the ion trap mass analyzer (normal scan rate) using collision induced dissociation (CID) with a normalized collision energy of 35 and an isolation window of 2.0 *m/z*. Maximum injection times of 50 ms were defined for both MS and MS/MS scans. AGC values were set to $1 \times 10^6$ for MS and $3 \times 10^4$ for MS/MS. MS data were collected in profile mode and MS/MS data were collected in centroid mode. For DIA, a full scan MS spectrum (*m/z* 300–1,100) was acquired in the Orbitrap with a resolution of 120,000 (at 200 *m/z*) and an AGC target of $5 \times 10^5$ or in the ion trap with an AGC target of $3 \times 10^4$. MS/MS was performed with an AGC target of $3 \times 10^4$ using an injection time limit of 30 or 60 ms[68]. All acquisitions were performed in positive mode polarity.

Data analysis was performed using our in-house software EpiProfile with a 10-ppm tolerance for extracting peak areas from raw files[45]. The relative abundances of acyl-PTMs were calculated by dividing the intensity the modified peptide by the sum of all modified and unmodified peptides sharing the same sequence, across all detectable charge states. For isobaric peptides (e.g., K$_{PTM}$STGGKAPR and KSTGGK$_{PTM}$APR), the relative abundances were estimated by extracting the area under the curve of unique fragment ions. DDA was used to confirm peptide elution time by performing Mascot database searching (via Proteome Discoverer v1.4), while DIA was used to integrate the extracted ion chromatogram with the Skyline software and EpiProfile.

**Generation of stable isotope labeled coenzyme A analogs**. Acyl-CoA labeled standards were generated using the stable isotope labeling by essential nutrients in cell culture (SILEC) protocol (70). Hepa1c1c7 cells were culture in modified DMEM media with 10% charcoal-stripped FBS where calcium pantothenate had been replaced with 1 mg per L $^{13}C_3^{15}N_1$-pantothenate for at least 7 generations. On the final passage, culture conditions for the Hepa1c1c7 SILEC labeling was modified by the addition of fresh SILEC media pH = 6.7 containing 1 mM disodium glutarate (BOC Sciences) and 1 mM potassium crotonate (Pfaltz & Bauer) for 1 h before harvest. This enriched acyl-CoA internal standard library was then mixed 1:1 with SILEC prepared from yeast[52] to increase the levels of more

common acyl-CoAs to reflect cellular levels. Yeast SILEC growth media was prepared with 200 μg biotin, 200 μg folic acid, 200 mg inositol, 40 mg niacin, 20 mg p-aminobenzoic acid, 40 mg pyridoxine-HCl, 20 mg riboflavin, 40 mg thiamine HCl, 20 g dextrose, 400 μg [$^{13}C_3^{15}N_1$]-pantothenate, 2.0 g Drop-out Mix Complete w/o Yeast Nitrogen Base, 1.7 g Yeast Nitrogen Base w/o AA & w/o AS, w/o Vitamins, and 5.0 g ammonium sulfate dissolved in 1 L of distilled $H_2O$. The vitamins and [$^{13}C_3^{15}N_1$]-pantothenate were filter sterilized and the dextrose, drop-out mix, yeast nitrogen base mix, and ammonium sulfate were autoclaved. To confirm pantothenate auxotrophy, agar plates were prepared, with one batch omitting pantothenate from the growth media. The plates were inoculated with *pan6Δ* yeast, and then incubated at 37 °C for 24 h. After confirming auxotrophy, 1 L of media was inoculated with *pan6Δ S. cerevisiae* and incubated at 30 °C while agitating overnight with 500 mL in two 2 L Erlenmeyer flasks covered loosely with aluminum foil. After approximately 31 h from the onset of the culture, the yeast cells were removed from the incubator, divided into 50 mL aliquots, and pelleted at 500×*g*. The cells were resuspended in ice-cold 10% TCA extraction, respectively. The cells were pulse sonicated for 30 half-second pulses, on ice. Samples were spun at 16,000×*g* for 10 min at 4 °C to remove unbroken cells and debris. The final supernatant was transferred to a separate tube and stored at −80 °C until use as the SILEC internal standard.

**Acyl-CoAs metabolites extraction and quantitation**. HeLa and myogenic cells collected from two confluent 15-cm plates (roughly $10–30 \times 10^6$ cells) were washed twice with PBS, scraped into a 15 mL conical tube and pelleted by centrifugation at 800×*g* for 3 min at 4 °C. Acyl-CoA metabolites were extracted following procedures already described[69]. In brief, pelleted cells were metabolically quenched and resuspended in 0.9 mL of ice-cold 10% TCA solution along with 0.1 mL of SILEC CoA internal standards, and sonicated with a probe tip sonicator for 30 s on ice with a 1 pulse per 2 s rate. Cell lysates were centrifuged at 14,000×*g* for 10 min at 4 °C to precipitate cellular debris and the supernatant containing the metabolites were kept on ice. Standards for calibration curves were prepared from commercially available acyl-CoAs (acetyl-, crotonyl-, malonyl-, propionyl-, butyryl-, glutaryl- and β-hydroxybutyryl CoA; —Sigma-Aldrich) as a master mix of 1 mM and diluted from 1 μM to 10 nM in 10% TCA. Succinyl-CoA was run as a separate standard curve due to lower purity of the standard used, but otherwise prepared identically. 50 μL of each standard solution was mixed with 0.85 mL of 10% TCA and spiked with 0.1 mL of SILEC CoA internal standard. Standard calibration curve samples were also subjected to sonication and solid phase extraction (SPE) in the same manner as the experimental samples to account for matrix effects, sample losses and analyte stability. Samples and standards were purified using SPE cartridges (Oasis HLB 10 mg) that were conditioned with 1 mL of methanol and equilibrated with 1 mL of water utilizing a vacuum manifold. Acid-extracted supernatants were loaded onto the cartridges and washed with 1 mL of water. Acyl-CoA metabolites were eluted with 1 mL of 25 mM ammonium acetate in methanol and dried under nitrogen. Samples were resuspended in 50 μL of 5% (w/v) 5-sulfosalicyilic acid and 10 μL injections were analyzed for acyl-CoAs by LC-HRMS and LC-MS/HRMS.

**Data availability**. The authors declare that all data supporting the findings of this study are available within the article and its supplementary information files or from the corresponding author upon reasonable request. All mass spectrometry raw files have been deposited in Chorus (https://chorusproject.org) under the project number 1376.

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

## Acknowledgements

We gratefully acknowledge Dr. Ben Black from the University of Pennsylvania for supplying the recombinant histones and Dr. Andrew Andrews from Fox Chase Cancer Center for supplying the recombinant enzymes CBP and p300. This work was supported by NIH grants AI118891, GM110174, CA196539 and AG031862 to B.A.G.; GM101664 to T.G.K.; K22ES26235, R21HD087866, R03CA211820 and a Pennsylvania Department of Health CURE grant to N.W.S.; R01 GM060293, R35 GM118090 and P01 AG031862 to R.M.; the NIH training grant fellowship T32GM008275 to D.M.M., and the NIH grant GM110174-S1 to M.C.

## Author contributions

J.S. performed most of the experiments, analyzed the data and wrote the manuscript; S.S. contributed to data analysis, made technical and intellectual contributions and helped writing the manuscript. Z.-F.Y. developed the software for analysis of histone marks and aided in the analysis of the histone data. M.C. performed the competition assays and the analysis of the ionization efficiencies of the differentially acylated peptide of histone H3. N.V.B. provided the myogenic cells; D.M.M. designed and performed initial experiments for CoA analysis and contributed to manuscript writing. B.J.K. and T.G.K. prepared the nucleosomal figures; G.A.B., C.E.M., and R.S.M. expressed and purified the recombinant HATs; N.W.S. performed the CoA sample preparation and data analysis and contributed to editing the manuscript; R.M. provided the recombinant enzymes and was available for helpful discussions and B.A.G. conceived, designed and supervised the study. All authors discussed the results and commented on the manuscript.

## Additional information

**Competing interests:** The authors declare no competing financial interests.

