## [Peer Review File · Nature Communications]

Reviewers' comments:

Reviewer #1 (Remarks to the Author):

The authors' provide a well-written introduction to the field of histone acylations and aim to answer important questions related to relative abundance and enzymatic versus non-enzymatic regulation. These are timely and important questions to address and will be of interest to chromatin biologists and those interested in the growing interface between metabolism and epigenetic mechanisms. This manuscript provides a series of much needed experiments from a group of respected scientists well-suited to address these issues. A major strength of this study is the comprehensive approach taken allowing for integration of various data sets from test tube reactions to in nucleo to differentiating cells. The major weakness of this study is the lack of nuance in the interpretation of results and lack of integration of results across experiment type. Conclusions drawn from individual experiments seem to be made in isolation of the other data in the manuscript. There is no attempt to synthesize the in vitro and cellular data. The in vitro experiments, while more comprehensive than previous publications, are not novel. The novelty in this manuscript is in the cellular and in nucleo experiments, yet only a superficial analysis of these data is attempted. Furthermore, the study suffers from over-interpretation of several in vitro assays and failure to raise critical caveats. Overall, the experiments and data are generally high quality, but the interpretation and analysis are shallow in comparison and in some instances unacceptable for publication. Either a major rewrite with a more accurate analysis of data and mention of caveats or generation of new data to support the claims made will be necessary for publication.

Specific Points:

1. The use of 500uM acyl-CoA to measure non-enzymatic acylation makes it difficult to interpret. Whole cell concentrations of acetyl-CoA are estimated to be ~1-10uM (Lee et al, Cancer Cell 2014), and nuclear concentrations will be significantly lower. Given the data presented in Figure 5 showing that other acyl-CoAs are dramatically less abundant, does this not introduce a major caveat to in vitro experiments conducted? The lower abundance acyl-CoAs are ~100-fold less abundant than acetyl-CoA. If we estimate 5uM for acetyl-CoA then you are using ~10,000x physiological concentrations of acyl-CoAs. This suggests that the non-enzymatic acylations observed are a dramatic exaggeration of what might be expected in the cell. This should be mentioned in the manuscript and introduced as a caveat in the interpretation of these data.
2. The data presented in figure 3 comparing enzymatic versus non-enzymatic acylations were conducted using only GCN5, an enzyme already reported to have high structural selectivity for acetyl-CoA and exclusion of other acyl-CoAs (Ringel and Wolberger, Acta Crystl. 2016). These data are then extrapolated to make generalized statements about how non-enzymatic a particular acylation is (figure 3b). Conclusions should be narrowed to statements about GCN5.
3. The final paragraph in the section "Histones are non-enzymatically acylated in vitro" needs to be rewritten to make these statements specific for GCN5 or analysis to support the general claims made need to be performed. For example, there is no suggestion in these data that "most acyl modifications identified on histones in vivo may be occurring by non-enzymatic mechanisms." The only suggestion supported by data presented in Figure 3 is that acyl-CoAs in general have non-enzymatic activity (including acetyl-CoA) and GCN5 has a strong preference for acetyl-CoA when compared to longer chain acyl-CoAs. The high concentrations of acyl-CoA used also makes extrapolation to in vivo statements inappropriate. This also applies to several general statements made in the discussion not supported by analysis of data.
4. The observation that crotonyl-CoA seems to have the lowest level of non-enzymatic transfer of all the tested acyl-CoAs seems to be the most novel aspect of figure 3 yet goes unmentioned by the authors.
5. The competition assays represented in figure 4 are a great idea in principle, but the concentrations of acetyl-CoA and acyl-CoAs make the data difficult to interpret. While it is fair to

use high concentration of acyl-CoAs in figure 1 to measure the potential maximum of a given HATs ability to utilize a given acyl-CoA, that is not the case for these competitive reactions. The optimal experiment would have taken into account the K_m of the various enzymes and adjusted the total concentration of acyl-CoA to be well below that concentration. The K_m of HATs ranges from 1-10 μ M (Albaugh et al 2011). Under these reaction conditions with 250 μ M of each acyl-CoA, these enzymes are completely saturated with acetyl-CoA. An only slightly higher affinity for acetyl-CoA would be expected to result in a complete usage of acetyl-CoA over the lower affinity acyl-CoAs. This is not the case within the cell where Acetyl-CoA concentrations are at or below the K_m for HATs. Again, the authors do not consider these caveats in their analysis.

6. The linear correlation analysis of acyl-CoA concentration and PTM abundance performed in figures 5e-g is inappropriate. The concentration of acetyl-CoA is so much greater than the rest of the acyl-CoAs that this type of statistical analysis is uninformative. It may be useful to present the data without acetyl-CoA to better visualize and evaluate the correlation for the lower-abundance acylations.

7. The increase in Kcr upon the depletion of acetyl-CoA and other acyl-CoAs observed in figure 5b can be explained by a model where some HATs have the capacity to utilize crotonyl-CoA in non-saturating conditions. In this model, even if acetyl-CoA is a more effective substrate, the loss of acetyl-CoA will lead to greater levels of Kcr.

8. While the in nucleo experiments are a powerful system to experimentally manipulate the concentrations of acyl-CoAs the analysis of results are cursory at best. How do the in nucleo results compare to the results of the in vitro experiments? What can be said about the responsiveness or fold change of the different acylations to increases in acyl-CoA concentration? What can be said about site utilization of the different acylations?

Minor comments:

1. Figure 2 seems out of place and could easily be supplemental. The figure is also misleading since the substrates used in the reactions are not mononucleosomes but unstructured histone monomers. Monomeric histones in solution do not have native structures.

2. In the final paragraph of section "Relative abundance of histone acyl modifications in mammalian cells," the introduction of results from a previous ACLY knockdown is awkward. There needs to be a little bit more context for the reader to digest this piece of information and integrate it with results presented in figure 5. Also, why would this result suggest that the deposition of acetylation plays an important role in myoblast differentiation? It actually suggests that loss of acetyl-CoA and potentially an increase in other acylations plays a positive role in myoblast differentiation.

3. In the discussion when making statements about acyl-CoA utilization by HATs cite (Ringel and Wolberger, 2016) and (Kaczmarek, et al 2016).

Reviewer #2 (Remarks to the Author):

In this study, the authors show evidence for the relative abundance of various histone acyl modifications from cultured cells, and show that their relative abundance correlates with the cellular levels of the corresponding acyl-CoA. They assessed in vitro the ability of seven histone acetyltransferases (HATs) to catalyze acyl addition to histones. They find that recombinant HATs are generally less active on these alternative substrates. Using isolated nuclei, the authors observed a dose-dependent increase in acyl PTM abundances when acyl CoAs were added to these samples. To date, this study documents the most complete profiling of the lower abundance acyl PTMs reported on histones. The authors conclude that the concentration of acyl-CoAs affects histone acyl-PTM abundances and that enzymatic and non-enzymatic mechanisms may be involved. This study contains important information of interest to investigators in the chromatin PTM field, however, there are some significant concerns that should be addressed.

- 1.) The acyl-transfer reactions should be performed with standard steady state kinetic approaches where [acyl-CoA] is varied, and saturation curves are fitted to obtain actual constants like V_{max} and K_m (for the recombinant HATs). The competition experiment is not a proper replacement for the more rigorous kinetics that yield constants that can be compared across multiple labs and publications, and importantly when compared with their own data using isolated nuclei. These experiments are not difficult. Acyl-CoA saturation curves should also be performed in the nuclei supplementation experiment. Not only can these results be compared to the isolated enzyme values, but the ability to saturate in the nuclei in vitro experiment might suggest an enzyme catalyzed reaction, perhaps revealing the existence of an enzyme that is not represented in the recombinant group. Regardless, it will be informative (and potentially supportive) toward the main conclusions of this work.
- 2.) It is important to note that both enzyme and non-enzyme catalyzed reactions increase the rate as a function of substrate. The ability to saturate enzyme is evidence for its existence.
- 3.) The authors need to be clear when they are speculating and when they are presenting observations: For example, the following sentence in the abstract is not well supported by the presented data: "They can use most acyl-CoAs, larger molecules such as glutaryl-CoA and succinyl-CoA are mostly added to lysine residues by non-enzymatic reactions." There are several examples of such statements that are an over-interpretation of the presented results.
- 4.) It appears that the authors didn't account for ionization efficiencies when reporting relative stoichiometry (which is termed "relative abundance"). In the experiment related to figure 5, the authors are measuring relative abundance of different acyl modifications of peptides. How do the authors control for different ionization efficiencies caused by drastically different chemical species: acetyl, propionyl, butyryl vs. succinyl, malonyl, glutaryl vs crotonyl, and can these modified peptides actually be compared directly? Do the authors correct for different ionization efficiencies as is done in Lin et al., 2014 MCP 13, 2450?
- 5.) In figure 3A, the authors present a bar graph showing GCN5 enzymatic activity towards histone H3 K-sites as well as nonenzymatic activity towards the same sites. Is the GNC5 activity that is reported for Ksuc, Kmal, Kbhb, Kglu, and Kcr, actually just non-enzymatic acylation caused by the concentration of acyl-CoA used in the assay? Do the authors subtract the background activity from a no enzyme control experiment?
- 6.) Figure 1E could be better represented to show the low abundant modifications. This low abundant data is not clearly visible and perhaps showing a different graphic will help illustrate these important points.
- 7.) Figure 4B and 4C seem to be representing the exact same data. The authors might consider showing only one plot such as the heatmap.
- 8.) In Figure 4D, instead of plotting a single linear regression for a single enzyme out of the seven, the authors should consider plotting the linear regression for all enzymes along with their R^2 values. If the data matches that for GCN5, it would really drive home the point to readers.
- 9.) In Figure 5E, F, & G, it would be helpful to show a plot zoomed into the low abundant modifications.

Reviewer #1 (Remarks to the Author):

The authors' provide a well-written introduction to the field of histone acylations and aim to answer important questions related to relative abundance and enzymatic versus non-enzymatic regulation. These are timely and important questions to address and will be of interest to chromatin biologists and those interested in the growing interface between metabolism and epigenetic mechanisms. This manuscript provides a series of much needed experiments from a group of respected scientists well-suited to address these issues. A major strength of this study is the comprehensive approach taken allowing for integration of various data sets from test tube reactions to in nucleo to differentiating cells. The major weakness of this study is the lack of nuance in the interpretation of results and lack of integration of results across experiment type. Conclusions drawn from individual experiments seem to be made in isolation of the other data in the manuscript. There is no attempt to synthesize the in vitro and cellular data. The in vitro experiments, while more comprehensive than previous publications, are not novel. The novelty in this manuscript is in the cellular and in nucleo experiments, yet only a superficial analysis of these data is attempted. Furthermore, the study suffers from over-interpretation of several in vitro assays and failure to raise critical caveats. Overall, the experiments and data are generally high quality, but the interpretation and analysis are shallow in comparison and in some instances unacceptable for publication. Either a major rewrite with a more accurate analysis of data and mention of caveats or generation of new data to support the claims made will be necessary for publication.

We thank the reviewer for his/her detailed analysis of our work. We really appreciate that the experiments we performed were considered as "much needed". We agree that in some parts we exceeded in drawing conclusions based on our data. We hope the reviewer will find this point-by-point reply appropriate and the new version of the manuscript satisfactory.

Specific Points:

1. The use of 500uM acyl-CoA to measure non-enzymatic acylation makes it difficult to interpret. Whole cell concentrations of acetyl-CoA are estimated to be ~1-10uM (Lee et al, Cancer Cell 2014), and nuclear concentrations will be significantly lower. Given the data presented in Figure 5 showing that other acyl-CoAs are dramatically less abundant, does this not introduce a major caveat to in vitro experiments conducted? The lower abundance acyl-CoAs are ~100-fold less abundant than acetyl-CoA. If we estimate 5uM for acetyl-CoA then you are using ~10,000x physiological concentrations of acyl-CoAs. This suggests that the non-enzymatic acylations observed are a dramatic exaggeration of what might be expected in the cell. This should be mentioned in the manuscript and introduced as a caveat in the interpretation of these data.

We agree with the reviewer's observation that the concentrations of acyl-CoAs used in our *in vitro* assays were above the known concentrations of acyl-CoAs in cells. It has been previously shown by Olia et al. that non-enzymatic histone acetylation can occur even at

physiological concentrations of acetyl-CoA *in vitro* (Olia AS et al. *Nonenzymatic Protein Acetylation Detected by NAPPA Protein Arrays*, ACS Chem Biol. 2015). Nonetheless, we acknowledge that our data cannot be directly extrapolated to *in vivo* systems. As such, we have re-written the results of the section “Histones are non-enzymatically acylated *in vitro*” emphasizing that while our data indicates that all acyl-CoAs intermediates evaluated are chemically reactive *in vitro*, our study is not a fair comparison to conditions in the cell (below in italic). [Page 8]

[...] we observed a trend in which the ratio of enzymatic/chemical reactivity of acyl groups is inversely correlated with the size of the side chain, supporting the idea that most known HATs are unable to catalyze the acylation of histones using larger acyl-CoA donors (39, 48). When looking individually at non-enzymatic acylations in histones, crotonyl-CoA showed the lowest acylation levels through chemical catalysis (Fig. 2c, 2d). As expected, crotonyl-CoA presents a lower chemical reactivity towards lysine nucleophiles due to the resonance properties of the beta unsaturated C-C bond (49).

The correlation between sites that were prone to chemical acylation in this study and sites identified in *in vivo* studies, together with the fact that enzymes with distinctive acyltransferase activities have not been identified in any cellular compartment (50), suggest that histone acyl modifications in cells could be the result of non-enzymatic mechanisms after direct exposure to intrinsically reactive acyl-CoA metabolites. Although the concentrations of acyl-CoAs used in this experiment were far above the known physiological concentrations of CoA derivatives in whole cells (51), it has been previously demonstrated that protein lysine acylation, can also occur at physiological acyl-CoA concentrations *in vitro* (48, 52). Nevertheless, even though we observed a pronounced chemical reactivity by CoA-metabolites *in vitro*, our study does not represent a suitable extrapolation for the reactivity of these intermediates in cells. As such, our data cannot rule out the possibility of the existence of enzymes that can catalyse these marks *in vivo*.

2. The data presented in figure 3 comparing enzymatic versus non-enzymatic acylations were conducted using only GCN5, an enzyme already reported to have high structural selectivity for acetyl-CoA and exclusion of other acyl-CoAs (Ringel and Wolberger, Acta Crystl. 2016). These data are then extrapolated to make generalized statements about how non-enzymatic a particular acylation is (figure 3b). Conclusions should be narrowed to statements about GCN5.

That was not clear in our manuscript. Figure 3a (now Figure 2a) was indeed prepared using only data from GCN5. However, figures 3b – 3d (now figures 2b – 2d) were prepared using the average contribution of all enzymatic (average of all enzymes evaluated) and non-enzymatic (average of all experiments carried out in the absence of enzymes) reactions as stated in the legend of the figure. We have re-written the section clarifying that figure 3b

represents the contribution of all enzymatic and non-enzymatic reactions evaluated in this study (below). [Page 29]

Figure 2. Non-enzymatic vs. enzymatic acylation of histones in vitro. (a) Comparison of non-enzymatic versus GCN5-catalyzed acylations on histone H3. The y-axis (arbitrary units) represents the sum of the relative abundances of all enzymatically and non-enzymatically acylated peptides from histones H3. We can observe that the contributions for Ksuc, Kmal, Kbhb, Kglu, and Kcr in the experiments with GCN5 were mostly the result of non-enzymatic acylations (b) Stacked column representation of non-enzymatic reactivity divided by enzymatic reactivity of the eight acyl-CoA donors on histone H3 (left) and histone H4 (right). The fractional reactivity represents the ratio of PTM intensity in presence of all eight enzymes vs. PTM intensity in absence of enzymes, i.e. 0.5 corresponds to identical intensities with and without enzyme. For instance, crotonylation is an overall low abundance PTM, although the majority detected on histone peptides is the result of an enzymatic catalysis.

3. The final paragraph in the section “Histones are non-enzymatically acylated *in vitro*” needs to be rewritten to make these statements specific for GCN5 or analysis to support the general claims made need to be performed. For example, there is no suggestion in these data that “most acyl modifications identified on histones *in vivo* may be occurring by non-enzymatic mechanisms.” The only suggestion supported by data presented in Figure 2 is that acyl-CoAs in general have non-enzymatic activity (including acetyl-CoA) and GCN5 has a strong preference for acetyl-CoA when compared to longer chain acyl-CoAs. The high concentrations of acyl-CoA used also makes extrapolation to *in vivo* statements inappropriate. This also applies to several general statements made in the discussion not supported by analysis of data.

After the edits thanks to the comment #2 we now think this issue has been addressed. [Page 29].

4. The observation that crotonyl-CoA seems to have the lowest level of non-enzymatic transfer of all the tested acyl-CoAs seems to be the most novel aspect of figure 3 yet goes unmentioned by the authors.

We appreciate the observation. We have included a paragraph under “Histones are non-enzymatically acylated *in vitro*” highlighting the low reactivity of crotonyl-CoA due to its low intrinsic energy (below in *italic*) [Page 8].

We then estimated which acylations are more likely to occur through enzymatic or non-enzymatic reactivity in vitro. As seen in figure 2b, the average contribution for acetylation, propionylation and butyrylation marks in the presence of all HATs was more abundant in in vitro experiments, whereas acidic acyl modifications (malonylation, succinylation and glutarylation) and β -hydroxybutyrylation occurred to a greater extent through non-enzymatic mechanisms in both histone H3 and H4. Again, we observed a trend in which the ratio of enzymatic/chemical reactivity of acyl groups is inversely correlated with the size of the side chain, supporting the idea that most known HATs are unable to catalyze the acylation of histones using larger acyl-CoA donors (references). When looking individually at non-enzymatic acylations in histones, crotonyl-CoA showed the lowest acylation levels through chemical catalysis (Fig. 2c, 2d). As expected, crotonyl-CoA presents a lower chemical reactivity towards lysine nucleophiles due to the resonance properties of the beta unsaturated C-C bond (49).

5. The competition assays represented in figure 4 are a great idea in principle, but the concentrations of acetyl-CoA and acyl-CoAs make the data difficult to interpret. While it is fair to use high concentration of acyl-CoAs in figure 1 to measure the potential maximum of a given HATs ability to utilize a given acyl-CoA, that is not the case for these competitive reactions. The optimal experiment would have taken into account the K_m of the various enzymes and adjusted the total concentration of acyl-CoA to be well below that concentration. The K_m of HATs ranges from 1-10 μ M (Albaugh et al 2011). Under these reaction conditions with 250 μ M of each acyl-CoA, these enzymes are completely saturated with acetyl-CoA. An only slightly higher affinity for acetyl-CoA would be expected to result in a complete usage of acetyl-CoA over the lower affinity acyl-CoAs. This is not the case within the cell where Acetyl-CoA concentrations are at or below the K_m for HATs. Again, the authors do not consider these caveats in their analysis.

The reviewer makes a very good point. We repeated the experiment utilizing physiological concentrations of acyl-CoAs (10 μ M of acetyl-Coa + 10 μ M of competing CoA) and our results showed that all enzymes mostly prefer acetyl-CoA as we observed before. Indeed, this experiment was very helpful to make our point more clear, as the new results showed an even better inverse correlation between the HATs preference for the competing cofactor and the increasing molecular weight of the acyl donor side chain. We have modified figure 3 to show the correlation for an enzyme representative of each HAT family (GCN5, p300 and MOF) (below). [Page 30]

FIGURE CAPTION –*In vitro* acetylation competition assay. (a) Schematic representation of the *in vitro* competition acetylation assay. (b) Heat map displaying *in vitro* acylation specificities of HATs during acyl-CoA competition assays. Different HATs were assayed against histones H3 or H4 as specified in the table headers in the presence of equimolar concentrations of acetyl-CoA and a competing acyl-CoA donor. (c) Correlation between the molecular weight and the acylation preference for different acyl-donors displayed for the HATs GCN5 on histone H3, (d) p300 on histone H3 and (e) MOF on histone H4. For each modification molecular weight is indicated. Results show that the preference for acetyl-CoA over the other acyl donor tightly correlates with the molecular weight of the acyl donor. Crotonylation was not included in the correlations, as its molecular weight did not correlate well with the preference of the enzymes over acetyl-CoA. All graphs are shown as the average of log2 ratios between the relative abundances of all acetylated peptides and the relative abundances of the corresponding competing acylated peptide.

6. The linear correlation analysis of acyl-CoA concentration and PTM abundance performed in figures 5e-g is inappropriate. The concentration of acetyl-CoA is so much greater than the rest of the acyl-CoAs that this type of statistical analysis is uninformative. It may be useful to present the data without acetyl-CoA to better visualize and evaluate the correlation for the lower-abundance acylations.

Figures 5e-5g (now figures 4e-4g) have been edited showing zoom in graphs with the linear correlation of the data without the values for acetyl-CoA. [Page 31]

*FIGURE CAPTION – (e-g) Global histone acylation levels correlation with intracellular concentrations of acyl-CoA metabolic intermediates in (e) HeLa cells, (f) myoblasts and (g) myotubes. Correlations are calculated between the normalized net abundances of all detectable acylated peptides in H3 and H4 and the global concentrations CoA metabolites in pmol/cell. All results are shown as the average of 3 biological replicates. Summary of p-values is as follows, * ($p \leq 0.05$), ** ($p \leq 0.01$), and *** ($p \leq 0.001$).*

7. The increase in Kcr upon the depletion of acetyl-CoA and other acyl-CoAs observed in figure 5b can be explained by a model where some HATs have the capacity to utilize crotonyl-CoA in non-saturating conditions. In this model, even if acetyl-CoA is a more effective substrate, the loss of acetyl-CoA will lead to greater levels of Kcr.

We appreciate the suggestion by the reviewer. In this manuscript, we avoided suggesting possible mechanisms involved in the increase of histone crotonylation observed during myoblast differentiation. As we mentioned in the discussion, this subject is currently being written up after a more detailed investigation carried out in the Garcia Lab.

8. While the *in nucleo* experiments that have been used in the chromatin biology field for decades are a powerful system to experimentally manipulate the concentrations of acyl-CoAs the analysis of results are cursory at best. How do the *in nucleo* results compare to the results of the *in vitro* experiments? What can be said about the responsiveness or fold change of the different acylations to increases in acyl-CoA concentration? What can be said about site utilization of the different acylations?

We agree with the reviewer that our description of the results was not sufficiently detailed. Indeed, comparing our *in vitro* results to our *in nucleo* results was not straightforward, as the responsiveness we see *in nucleo* could not be attributed to an enzymatic or non-enzymatic mechanism. The reason we performed this intermediate experiment was to prove that the observed relationship between the abundances of acyl-PTMs *in vivo* with the levels of metabolites would hold if we manipulated the system. As the results showed, we did observe an increase in the levels of histones acylations with the supplementation of artificial levels of acyl-CoAs. We have rewritten this section emphasizing more in the details and concerns raised by the reviewer and included a supplementary figure S7 showing the *in vitro* – *in nucleo* correlation for some sites (text and figure below). [Pages 13]

Acylation of histones in nucleo: Our *in vivo* and metabolomics studies strongly implicated metabolism in histone acylations. As the concentrations of intracellular metabolites are known to change in response to diet or physiological conditions, we next turned to the question of whether alterations in the levels of metabolites could affect the corresponding levels of histone acylations. To further explore this idea, we turned to an *in nucleo* system, a model where purified nuclei can be treated with artificial levels of metabolites. We isolated nuclei from HeLa cells under hypotonic conditions, treated with varying concentrations of eight different acyl-CoA donors and performed histone extraction, digestion and derivatization following standard procedures (Fig. 5a).

MS analysis revealed that histone acylations can be induced in a concentration-dependent manner. Specifically, by adding 1 or 5 μM of acyl-CoAs we induced an increase of the respective acylation on histone peptides (Fig. 5b). *In nucleo* conditions preserve the activities of chromatin modifying enzymes, representing a suitable model for the study of histone acylations in native chromatin. However, unlike our *in vitro* experiments, under *in nucleo* conditions it is not feasible to discriminate between enzymatic and non-enzymatic mechanisms. Therefore, we cannot fully attribute the increase in acylation to the utilization of metabolites by HATs, especially after the observed chemical reactivity of CoA donors at low concentrations in our competition experiments (Fig 3b). An *in nucleo* – *in vitro* Spearman's rank-order correlation analysis was run to determine the relationship between site occupancy in both experiments. We observed a good correlation for H3K9_{acyl} and H3K18_{acyl} sites (Fig. S7). However, the same correlation was not observed when performing the same analysis between sites that were non-enzymatically acylated in *in vitro* experiments with sites acylated during *in nucleo* experiments (Fig. S7). Although we could suggest that the mechanisms for the acylation of these sites are enzymatic in nature, our data cannot strongly support this argument.

Additionally, our *in nucleo* results showed that histones accommodate acylation in two ways; by simply increasing the modified state or by removing pre-existing modifications to maintain the same level of total modified form. For example, upon treatment with malonyl-CoA, levels of H3K56mal increased with almost no changes in the other modifications on that particular peptide (Fig. 5c). On the other hand, after treatment with increasing concentrations of propionyl-CoA, the levels of H3K14pr increased while the levels of H3K14ac showed a measurable decrease (Fig. 5d).

Taken together, our *in nucleo* studies demonstrated that modifications in chromatin are sensitive to changes in the concentrations of cellular metabolites, consistent with previous observations connecting the metabolic state of the cell with chromatin regulation (9-11).

Supplementary figure 7. In vitro – in nucleo site occupancy Spearman correlation analysis.

Graphs show the correlation between H3K9_{acyl} and H3K18_{acyl} site occupancy in in vitro and in nucleo experiments. (a) The data represent ranked-order relative abundances of these sites in in vitro experiments in the presence of HATs vs. their abundance in in nucleo experiments supplemented with 5 μ M acyl-CoAs, and (b) in vitro experiments in the absence of HATs vs. in nucleo experiments supplemented with 5 μ M acyl-CoAs. K_{bhb} marks were not included in the correlations. Data were displayed as ranks to overcome the issue of the large dynamic range between acetyl marks and all other acyl marks; Pearson correlation would produce high R^2 independently from the relative abundance of the low intensity acyl marks (data not shown).

Minor comments:

1. Figure 2 seems out of place and could easily be supplemental. The figure is also misleading since the substrates used in the reactions are not mononucleosomes but unstructured histone monomers. Monomeric histones in solution do not have native structures.

We have worked to make this figure clearer, and have also moved it to the Supplementary Material as the Reviewer suggested. We have revised the caption of the figure (below in italic).

Supplementary figure 1. Representative nucleosome structure depicting site specificities for enzymatic and non-enzymatic acylations. This figure only represents the mapping of the *in vitro* acylation sites on H3 and H4 if our experiments were carried out using folded histones. Because our experiments were performed in the context of free unfolded histones, this figure is not a realistic representation. (a-b) Crystal structure of the nucleosome (PDB ID: 1KX5) displaying HAT acylation specificities for lysine residues in histone (a) H3 and (B) H4 shown in a space-filling model. Preferential acylation states of each modified lysine are indicated by magenta and yellow colors for histone H3 and H4, respectively. DNA is light green. Lysine modifications are shown as: acetylation (ac), propionylation (pr), butyrylation (bu), crotonylation (cr), malonylation (mal), succinylation (suc), and glutarylation (glu). (c) Most prevalent sites for non-enzymatic acylation are mapped onto the surface representation of the nucleosome structure (PDB ID: 1KX5).

2. In the final paragraph of section “Relative abundance of histone acyl modifications in mammalian cells,” the introduction of results from a previous ACLY knockdown is awkward. There needs to be a little bit more context for the reader to digest this piece of information and integrate it with results presented in figure 5.

Also, why would this result suggest that the deposition of acetylation plays an important role in myoblast differentiation? It actually suggests that loss of acetyl-CoA and potentially an increase in other acylations plays a positive role in myoblast differentiation.

We thank the reviewer for this comment and have now revised the text accordingly (below in italic). [Page 11]

Our analysis of myoblasts showed dynamic changes in global histone acylation upon their fusion to form multinucleated myotube cells. We observed that the global levels of lysine acetylation, propionylation, butyrylation, malonylation, succinylation and glutarylation were significantly decreased upon myogenic differentiation, whereas the levels of lysine crotonylation were increased (Fig. 4b). The role of differential histone acylation in cellular

differentiation is poorly understood; however, various lines of evidence have suggested that nutrition and metabolism do play a key role in the differentiation of cells (58, 59). For example, a previous study investigating the role of carbon metabolism in the differentiation of myogenic cells demonstrated that siRNA knock down of ATP citrate lyase (ACLY) induces differentiation of mouse myoblasts (60). The same study also showed that the levels of histone acetylation in shAcl- treated cells were reduced, hypothesizing that the deposition of acetyl-CoA and in turn histone acetylation levels, play an important role in the differentiation of myoblasts. Because there is little evidence of how newly identified histones acylations may be implicated in the differentiation of cells and/or epigenetic regulations, we next sought to investigate whether the cellular concentrations of other acyl-CoA metabolites have a direct relationship in histone acylation levels.

3. In the discussion when making statements about acyl-CoA utilization by HATs cite (Ringel and Wolberger, 2016) and (Kaczmarska, et al 2016).

Thank you also for this accurate reading. We have now added the suggested references [Page 15].

Reviewer #2 (Remarks to the Author):

In this study, the authors show evidence for the relative abundance of various histone acyl modifications from cultured cells, and show that their relative abundance correlates with the cellular levels of the corresponding acyl-CoA. They assessed in vitro the ability of seven histone acetyltransferases (HATs) to catalyze acyl addition to histones. They find that recombinant HATs are generally less active on these alternative substrates. Using isolated nuclei, the authors observed a dose-dependent increase in acyl PTM abundances when acyl CoAs were added to these samples. To date, this study documents the most complete profiling of the lower abundance acyl PTMs reported on histones. The authors conclude that the concentration of acyl- CoAs affects histone acyl-PTM abundances and that enzymatic and non-enzymatic mechanisms may be involved. This study contains important information of interest to investigators in the chromatin PTM field, however, there are some significant concerns that should be addressed.

We thank the reviewer for the very detailed analysis of our manuscript. We have addressed the concerns raised. We admit we have a concern regarding the issue #1, due to the technical difficulties of such experiment, and the fact that the result would go beyond the goals of the current manuscript. We hope the reviewer will find our new version satisfactory nevertheless.

1.) The acyl-transfer reactions should be performed with standard steady state kinetic approaches where [acyl-CoA] is varied, and saturation curves are fitted to obtain actual constants like V_{max} and K_m (for the recombinant HATs). The competition experiment is not a proper replacement for the more rigorous kinetics that yield constants that can be compared across multiple labs and publications, and importantly when compared with their own data using isolated nuclei. These experiments are not difficult. Acyl-CoA saturation curves should also be performed in the nuclei supplementation experiment. Not only can these results be compared to the isolated enzyme values, but the ability to saturate in the nuclei in vitro experiment might suggest an enzyme catalyzed reaction, perhaps revealing the existence of an enzyme that is not represented in the recombinant group. Regardless, it will be informative (and potentially supportive) toward the main conclusions of this work.

We thank the reviewer for this comment and we agree. However, these experiments are prohibitive in terms of time and costs for mass spectrometry, and truthfully they are beyond the take home message and scope of our manuscript, which was mostly to show the relative abundances of histone acylations in mammalian cells for the first time and their correlation to the levels of acyl-CoA metabolites.

We agree that this type of analysis seems like a great follow up experiment to derive parameters such as K_m or V_{max} constants. Actually, our collaborators from the Marmorstein Lab do perform kinetic characterization of HATs utilizing a radioactivity-based assay that

measures the transfer of radiolabeled CoA substrates and the bulk incorporation of acylations into histone peptides or proteins. Implementing such an assay would be very costly for us because it requires all CoA substrates to be radiolabeled. Additionally, radiolabeled assays would not give us information regarding HATs site specificity, which was crucial to our findings. However, we definitely hope other scientists will take over our results to perform more targeted enzyme characterization in a lower throughput manner.

2.) It is important to note that both enzyme and non-enzyme catalyzed reactions increase the rate as a function of substrate. The ability to saturate enzyme is evidence for its existence.

We thank the reviewer for this comment and have now revised the text accordingly (below in italic) [Page 8].

The correlation between sites that were prone to chemical acylation in this study and sites identified in in vivo studies, together with the fact that enzymes with distinctive acyltransferase activities have not been identified in any cellular compartment (49), suggest that histone acyl modifications in cells could be the result of non-enzymatic mechanisms after direct exposure to intrinsically reactive acyl-CoA metabolites. Although the concentrations of acyl-CoAs used in this experiment were far above the known physiological concentrations of CoA derivatives in whole cells (50), it has been previously demonstrated that protein lysine acylation, especially acetylation, can also occur at physiological acetyl-CoA concentrations in vitro (51). Nevertheless, even though we observed a pronounced chemical reactivity by CoA-metabolites in vitro, our study does not represent a suitable extrapolation for the reactivity of these intermediates in cells. As such, our data cannot rule out the possibility of the existence of enzymes that can catalyse these marks in vivo.

3.) The authors need to be clear when they are speculating and when they are presenting observations: For example, the following sentence in the abstract is not well supported by the presented data: "They can use most acyl-CoAs, larger molecules such as glutaryl-CoA and succinyl-CoA are mostly added to lysine residues by non-enzymatic reactions." There are several examples of such statements that are an over-interpretation of the presented results.

We thank the reviewer for this comment. We have extensively revised the entire manuscript, being careful on this issue.

4.) It appears that the authors didn't account for ionization efficiencies when reporting relative stoichiometry (which is termed "relative abundance"). In the experiment related to figure 5, the authors are measuring relative abundance of different acyl modifications of peptides. How do the authors control for different ionization efficiencies caused by drastically different chemical species: acetyl, propionyl, butyryl vs. succinyl, malonyl, glutaryl vs crotonyl, and can these modified peptides actually be compared directly? Do the authors correct for different ionization efficiencies as is done in Lin et al., 2014 MCP 13, 2450?

That is an important issue. On purpose we refer to our quantitative values as "relative abundance". We do not think we are currently able to assess accurate stoichiometry of our modified peptides for the following reasons: (i) the only true way to account for these changes would be to create a synthetic peptide for every modified peptide, but this is obviously a prohibitive task; (ii) certain classes of modified peptides are actually very hard to synthesize; (iii) by using synthetic peptides we correct only for ionization efficiency, but there are other issues that could affect the proper estimation of PTM stoichiometry, e.g. biases in trypsin digestion in presence of certain modifications.

Nevertheless, we definitely value the importance of the issue raised. Therefore, we synthesized the peptide of histone H3 aa 18-26 (KQLATKAAR) with all eight potential modifications on the residue K18. All peptides were derivatized with d5 propionylation, so that they could be all mixed together for the analysis, including K18pr. We observed some difference in the ionization efficiency, although all within a maximum of 3 fold changes (Fig. (1)). This deviation was larger than previously observed between acetylation and methylation residues (Fig. (2), taken from Figure 6A in PMID: 25000943), but still within a range we consider acceptable. We included this information in supplementary material, and made clearer in the manuscript that relative abundance does not reflect accurate stoichiometry (below in *italic*). [Page 10]

Thus, to accurately detect and quantify histone acyl-PTMs in vivo using label-free approaches without applying immunoenrichment we used the retention time and mass shift

(1)

(2)

information from the *in vitro* experiments to optimize the MS acquisition and in-house quantification software. Spectra were acquired using data dependent acquisition (DDA) and data independent acquisition (DIA) (55). DDA was used to confirm peptide elution time by performing Mascot database searching (via Proteome Discoverer v1.4), while DIA was used to integrate the extracted ion chromatogram with the Skyline software and our in-house software EpiProfile (56). Here, we refer to our quantitative values as “relative abundances” as we are aware that differences in the ionization efficiencies of modified peptides or biases in trypsin digestion in the presence of certain modifications can affect the assessment of accurate PTM stoichiometry, which we have observed for the differentially acylated peptide H3 aa 18-26 (KQLATKAAR) (Fig. S4).

Supplementary figure 4. Ionization efficiency profiles for differentially acylated peptide H3 aa 18-26 (KQLATKAAR). Bar plots shows the peak areas of synthetic peptide KQLATKAAR with eight different acyl modifications on lysine K18. PTMs (Kac: lysine acetylation, Kcr: lysine crotonylation, Kbu: lysine butyrylation, Kmal: lysine malonylation, Ksu: lysine succinylation, Kglu: lysine glutarylation, and Kpr: lysine propionylation). 2µg of each modified peptide were mixed together, derivatized with d10-propionic anhydride and analyzed by LC-MS. We observed some differences in the ionization efficiency, although all within a maximum of 3 fold changes, but still within a range we consider acceptable. All results are shown as the average of 3 experiments.

5.) In figure 3A, the authors present a bar graph showing GCN5 enzymatic activity towards histone H3 K-sites as well as nonenzymatic activity towards the same sites. Is the GNC5 activity that is reported for Ksuc, Kmal, Kbhb, Kglu, and Kcr, actually just non-enzymatic acylation caused by the concentration of acyl-CoA used in the assay? Do the authors subtract the background activity from a no enzyme control experiment?

We thank the reviewer for this comment and have now revised the figure 3 (now figure 2) accordingly. In all our result for figure 1, the non-enzymatic acylation was subtracted using the respective negative control experiment with no enzyme as described in the legend of figure 1. However, in figure 2a, we wanted to contrast the contributions of enzymatic vs. non enzymatic acylations in the presence of the enzyme GCN5, therefore, we did not subtract the non-enzymatic contribution, but instead plotted it next to the contributions of GCN5 to the acylation of H3. Effectively, the figure now clearly shows the message we wanted to transmit, that is, that most of the Ksuc, Kmal, Kbhb, Kglu, and Kcr observed in GCN5 experiments, were the results of non-enzymatic mechanisms. We have also revised the legend for figure 2, explaining that while the plot in figure 2a was prepared only using the data of GCN5 and the respective negative controls, the rest of the figures 2b was prepared using the data of all enzymes and figures 2c-2d were prepared using only the data from negative controls [Page 29].

FIGURE CAPTION. Non-enzymatic vs. enzymatic acylation of histones *in vitro*. (a) Comparison of non-enzymatic versus GCN5-catalyzed acylations on histone H3. The y-axis (arbitrary units) represents the sum of the relative abundances of all enzymatically and non-enzymatically acylated peptides from histones H3. We can observe that the contributions for Ksuc, Kmal, Kbhb, Kglu, and Kcr in the experiments with GCN5 were mostly the result of non-enzymatic acylations (b) Stacked column representation of non-enzymatic reactivity divided by enzymatic reactivity of the eight acyl-CoA donors on histone H3 (left) and histone H4 (right). The fractional reactivity represents the ratio of PTM intensity in presence of all eight enzymes vs. PTM intensity in absence of enzymes, i.e. 0.5 corresponds to identical intensities with and without enzyme. For instance, crotonylation is an overall low abundance PTM, although the majority detected on histone peptides is the result of an enzymatic catalysis. (c-d) Bar plot showing the relative quantitation of non-enzymatically acylated sites on (c) histones H3 and (d) histone H4. All results are shown as the average of 3 independent experiments.

6.) Figure 1E could be better represented to show the low abundant modifications. This low abundant data is not clearly visible and perhaps showing a different graphic will help illustrate these important points.

We thank the reviewer for this comment and have now revised the Figure accordingly. We converted the graph into a bar plot, which is easier to interpret than radar plots. In addition, we scaled the perspective and made the bars transparent enough to provide a better overview of all relative abundances of the modifications [Page 28].

7.) Figure 4B and 4C seem to be representing the exact same data. The authors might consider showing only one plot such as the heatmap.

We thank the reviewer for this comment and have now removed Figure 4C.

8.) In Figure 4D, instead of plotting a single linear regression for a single enzyme out of the seven, the authors should consider plotting the linear regression for all enzymes along with their R2 values. If the data matches that for GCN5, it would really drive home the point to readers.

We appreciate the comments from the reviewer. Plotting the correlations for all seven enzymes would result in a very messy graph; therefore, we have now plotted the correlation for 3 enzymes, one for each HAT family (GCN5, p300 and MOF). Now that we have

removed figure 4C and added figures 4C-E we hope the reviewer finds figure 4 more appropriate and informative [Page 30].

FIGURE CAPTION –*In vitro* acetylation competition assay. (a) Schematic representation of the *in vitro* competition acetylation assay. (b) Heat map displaying *in vitro* acylation specificities of HATs during acyl-CoA competition assays. Different HATs were assayed

against histones H3 or H4 as specified in the table headers in the presence of equimolar concentrations of acetyl-CoA and a competing acyl-CoA donor. (c) Correlation between the molecular weight and the acylation preference for different acyl-donors displayed for the HATs GCN5 on histone H3, (d) p300 on histone H3 and (e) MOF on histone H4. For each modification molecular weight is indicated. Results show that the preference for acetyl-CoA over the other acyl donor tightly correlates with the molecular weight of the acyl donor. Crotonylation was not included in the correlations, as its molecular weight did not correlate well with the preference of the enzymes over acetyl-CoA. All graphs are shown as the average of log2 ratios between the relative abundances of all acetylated peptides and the relative abundances of the corresponding competing acylated peptide.

9.) In Figure 5E, F, & G, it would be helpful to show a plot zoomed into the low abundant modifications.

Figures 5e-5g (now figures 4e-4g) have been edited showing zoom in graphs with the linear correlation of the data without the values for acetyl-CoA [Page31].

*FIGURE CAPTION – (e-g) Global histone acylation levels correlation with intracellular concentrations of acyl-CoA metabolic intermediates in (e) HeLa cells, (f) myoblasts and (g) myotubes. Correlations are calculated between the normalized net abundances of all detectable acylated peptides in H3 and H4 and the global concentrations CoA metabolites in pmol/cell. Insets show zoom in graphs with the linear correlation of the data without the values for acetyl-CoA and Kac. All results are shown as the average of 3 biological replicates. Summary of p-values is as follows, * ($p \leq 0.05$), ** ($p \leq 0.01$), and *** ($p \leq 0.001$).*

REVIEWERS' COMMENTS:

Reviewer #1 (Remarks to the Author):

While the revised manuscript is improved and the reviewer appreciates the effort placed in generating new data to address some of the reviewers' concerns, the same overall critique still stands that "the interpretation and analysis are shallow ... and in some instances unacceptable for publication. A major rewrite with a more accurate analysis of data and mention of caveats ... will be necessary for publication." This reviewer does not feel that the authors' have addressed this overall concern. There are several examples where statements are made that are not fully supported by the data. Furthermore, there are statements made that are refuted by the authors' own data. The comments from the first revision still hold: The major weakness of this study is the lack of nuance in the interpretation of results and lack of integration of results across experiment type. Conclusions drawn from individual experiments seem to be made in isolation of the other data in the manuscript. Because this is a fairly complex subject with 8 different acylations and 6 different enzymes, specific and precise language is necessary for clarity. These are timely and important experiments. There are exciting data presented in this study that should be published, but the manuscript needs to be revised. Below are specific comments followed by minor comments.

Specific comments

Figure 1

The authors did not take the advice of both reviewers to more carefully interpret the *in vitro* assays. These are not straightforward results to interpret and many readers will take the word of the authors' without digging into supplemental data to understand the details. It is surprising that the authors do not attempt a more nuanced and contextualized analysis of their own data. It is particularly surprising given the collective expertise of the authors possess in the chromatin field.

A closer analysis of the data in supplementary table 1 shows that the data does not support the authors' generalized statements about the six HATs tested. While the data for GCN5, PCAF and CBP seem to support the authors' statements, data for MOF, TIP60 and p300 seem to contradict the reported "overall trend" to the degree that to talk of an "overall trend" is inappropriate. I will go through each in turn.

MOF seems to prefer propionyl-CoA as a cofactor relative to acetyl-CoA. The data show that MOF is able to fully propionylate (all 4 lysines) close to 80% of the H4 N-tail, whereas it only approaches 40% full acetylation. This is a very surprising and convincing result that directly contradicts the generalized statements made by the authors.

TIP60 seems to prefer But-CoA as a cofactor, followed by succ-CoA and then acetyl-CoA. In fact, by these assay results, it looks as though TIP60 is not a very good acetyl-transferase. Furthermore, these data demonstrate that TIP60 utilizes hib-CoA to catalyze H4K59hib, which is the most abundant single modification analyzed for TIP60. H4K59hib is not observed in the no-enzyme control, suggesting that this modification is enzyme-dependent. Again, these novel and highly relevant observations contradict the "overall trend," but go unreported by the authors.

As has been previously observed, p300 is the most promiscuous acyltransferase in terms of sites of modification and of acyl-CoA usage and the authors' data support this notion. Focusing on H3K18, the lysine that the authors report is the dominant site of p300 activity, the data show that p300 utilizes CrCoA and Bhb-CoA at ~20% the levels as they utilize acetyl-CoA. These are by no means insignificant activities. This is the first demonstration that p300 has the capacity to catalyze Kbhb and the Kcr results are corroborated by published data (Lui X et al. Cell Discovery 2017 and Sabari BR et al. Mol Cell 2015) yet these data go unmentioned by the authors.

The observed differences between the enzymes make the generalized analysis performed in Figures 1c, 1d, and 1e unreasonable and obfuscating. Furthermore, the heavy-handed interpretation of these analyses seems inappropriate. It is surprising that the authors allow so much incongruity with their statements to go unmentioned in the text. Why generate these rich data matrices if they go unanalyzed?

Figure 2:

The analysis of non-enzymatic acylation does not seem to take into account the 500uM concentration of acyl-CoAs used. As reviewer #2 pointed out, the rate of both enzymatic and non-enzymatic acylation will be dependent on substrate (acyl-CoA) concentration, but only the enzymatic reaction rate will saturate. By using excessive levels of acyl-CoA the authors are exaggerating the relative abundance of non-enzymatic acylation compared to enzymatic acylation. When trying to compare enzymatic versus non-enzymatic acylation, 500uM acyl-CoA will 1) underestimate the relative enzymatic contribution because these concentrations are much higher than measured K_m s of HATs and most likely beyond concentrations for V_{max} and 2) overestimate the non-enzymatic contribution as the rate of chemical acylation will not be throttled by an enzyme's V_{max} . This is insufficiently acknowledged in the revision. The authors' citation of a study demonstrating that non-enzymatic acylation can also occur at physiological concentrations is irrelevant to these concerns.

"supporting the idea that most known HATs are unable to catalyze the acylation of histones using larger acyl-CoA donors"

The authors' own data demonstrate that several HATs are "able" to catalyze larger acylations. Instead of "unable," the authors should write "HATs catalyze larger acylation less efficiently." This is just one example of how the strength of the conclusions does not match the data presented. The authors should be more precise with their language.

Figure 3

The reviewer appreciates that the authors redid the competition assays at a lower concentration of acyl-CoA, but it seems as though they misinterpreted the comments from the first revision. 10uM is still too high as it is well above the reported K_m of most HATs and at the extreme end of reported whole cell concentrations. The authors' calling 10uM "low concentration of acyl-CoAs" does not make it so. Data from the literature demonstrate that crotonyl-CoAs can compete with acetyl-CoA for p300 acyltransferase activity at 2-4uM acyl-CoA concentrations (Sabari et al Mol Cell 2015).

Also, the data for the competition assays are not provided. Relevant mass spec data should be provided as a supplementary table.

Figure 5

"However, the same correlation was not observed when performing the same analysis between sites that were non-enzymatically acylated in in vitro experiments with sites acylated during in nucleo experiments (Fig. S7)."

How does this not refute the relevance of your 500uM acyl-CoA in vitro experiments or the statements made referencing them?

Mass spec data should be provided as a supplementary table.

Discussion

"However, they performed very poorly at catalyzing the acylation of histones with charged, branched or planar acyl-CoA cofactors."

The authors' own data does not support this statement (see comments on Figure 1). The authors' should be more precise with their language.

"our data does not support recent hypotheses suggesting that some novel acyl-PTMs could be mediated by any of the HATs explored in this study in vivo."

The in vitro experiments, particularly the competition reactions, presented in this study have little relevance to the situation in vivo. In fact, the data from the authors' in vitro and in nucleo experiments refute this statement. The authors show increases in sites of acylation for which they demonstrate HATs have the capacity to catalyze. The authors even make the statement about how the in nucleo acylation matches more to the enzymatic residues than to the non-enzymatic residues, which they define by using known HATs. Furthermore, some of the HATs utilize pr-CoA and but-CoA to equivalent levels as ac-CoA. Are they not included in "some novel acyl-PTMs"? The authors' should be more precise with their language.

"Thus, our data supports emerging models suggesting that non-enzymatic chemical reactions are a major contributor to the landscape of lysine acylations in nuclear histones (65)."

The only experiment that comes close to recapitulating the situation in a live cell, the in nucleo experiments, does not support this statement. The only cell-based experiment presented, myotube differentiation, shows a change in histone acylation that cannot easily be explained by non-enzymatic mechanisms. The only support for such a statement is the heavy-handed interpretation of single time-point, single concentration in vitro reactions. It is unclear where the authors are generating the confidence to make such a generalized statement about all histone acylations. The authors' should be more precise with their language.

Interestingly, a good in vitro – in nucleo correlation was observed for specific sites in histone H3, suggesting the implication of HATs in the utilization of supplemented acyl-CoAs for the acylation of histones.

How does this not refute your non-enzymatic claims? How can you ignore this discrepancy in logic?

Minor comments

Page 4:

1) "The dearth of quantitative data for these non-canonical acyl histone PTMs has led to the hypothesis that they might be nothing more than metabolic noise, arising simply from the chemical reactivity of acyl-CoAs."

Loaded phrases like "nothing more than," "noise," "simply" don't seem necessary. The same statement can be made as "led to the hypothesis that they might arise due to the chemical reactivity of acyl-CoAs."

2) "This last possibility might force us to reappraise their biological relevance"

This statement is unnecessarily charged and doesn't seem appropriate to the scope of the paper.

Page 8: "However, in competition assays perform in presence..." should be "performed"

Page 11: "shACL-treated cells" should be "shACLY-treated cells"

Page 16: "Intrigingly" should be "Intriguingly"

Page 17: "sugesting the implication of HATs in the utilization of supplemented..." should be "suggesting"

Page 33: Figure 5d. The colors used to represent data for H3K14pr and H3K14ac are too similar. As these are two colors the reader is trying to distinguish in the graph, it would be useful if they were more distinct from one another.

Reviewer #2 (Remarks to the Author):

The authors have done an excellent job addressing the comments and concerns raised in the original review. These results will be of tremendous interest to investigators working in the proteomics, chromatin, and protein acylation fields. The manuscript has been greatly improved and I am supportive of publication. John Denu

Reviewer #1 (Remarks to the Author):

While the revised manuscript is improved and the reviewer appreciates the effort placed in generating new data to address some of the reviewers' concerns, the same overall critique still stands that "the interpretation and analysis are shallow ... and in some instances unacceptable for publication. A major rewrite with a more accurate analysis of data and mention of caveats ... will be necessary for publication." This reviewer does not feel that the authors' have addressed this overall concern. There are several examples where statements are made that are not fully supported by the data. Furthermore, there are statements made that are refuted by the authors' own data. The comments from the first revision still hold: The major weakness of this study is the lack of nuance in the interpretation of results and lack of integration of results across experiment type. Conclusions drawn from individual experiments seem to be made in isolation of the other data in the manuscript.

We appreciate the reviewer's comments. Honestly, we agree that *in vitro* and *in nucleo* experiments cannot exhaustively reveal univocal relationships between histone writers and modifications. Moreover, we extensively showed that these modifications are difficult to regulate, as they can occur even without enzymes. It is our intention to not present drastic conclusions, as the equilibrium of these low abundance PTMs is more complex than these assays can describe. However, we know the reviewer agrees with us that these observations are a fundamental first step towards (i) understanding the roles of these PTMs, (ii) providing an estimation of their abundance *in vivo*, and (iii) drawing connections between metabolism and acyl-PTMs. All this is very much needed, at least for the epigenetics scientific community, as it is currently unclear the preponderance of these PTMs and how they are deposited. We also know the reviewer understands this dataset is not of simple production. We realized our statements might have been approximate occasionally. By trying to not overestimate the outcome of single results we ended up sometimes making conclusions that were too superficial. We hope the reviewer will appreciate our new revision, where we put all our efforts towards avoiding this issue.

Because this is a fairly complex subject with 8 different acylations and 6 different enzymes, specific and precise language is necessary for clarity. These are timely and important experiments. There are exciting data presented in this study that should be published, but the manuscript needs to be revised. Below are specific comments followed by minor comments.

Specific Points:

1. Figure 1

The authors did not take the advice of both reviewers to more carefully interpret the *in vitro* assays. These are not straightforward results to interpret and many readers will take the word of the authors' without digging into supplemental data to understand the details. It is surprising that the authors do not attempt a more nuanced and contextualized analysis of their own data. It is particularly surprising given the collective expertise of the authors possess in the chromatin field.

A closer analysis of the data in supplementary table 1 shows that the data does not support the authors' generalized statements about the six HATs tested. While the data for GCN5, PCAF and CBP seem to support the authors' statements, data for MOF, TIP60 and p300 seem to contradict the reported "overall trend" to the degree that to talk of an "overall trend" is inappropriate. I will go through each in turn.

MOF seems to prefer propionyl-CoA as a cofactor relative to acetyl-CoA. The data show that MOF is able to fully propionylate (all 4 lysines) close to 80% of the H4 N-tail, whereas it only approaches 40% full acetylation. This is a very surprising and convincing result that directly contradicts the generalized statements made by the authors.

TIP60 seems to prefer But-CoA as a cofactor, followed by succ-CoA and then acetyl-CoA. In fact, by these assay results, it looks as though TIP60 is not a very good acetyl-transferase. Furthermore, these data demonstrate that TIP60 utilizes hib-CoA to catalyze H4K59hib, which is the most abundant single modification analyzed for TIP60. H4K59hib is not observed in the no-enzyme control, suggesting that this modification is enzyme-dependent. Again, these novel and highly relevant observations contradict the "overall trend," but go unreported by the authors.

As has been previously observed, p300 is the most promiscuous acyltransferase in terms of sites of modification and of acyl-CoA usage and the authors' data support this notion. Focusing on H3K18, the lysine that the authors report is the dominant site of p300 activity, the data show that p300 utilizes CrCoA and Bhb-CoA at ~20% the levels as they utilize acetyl-CoA. These are by no means insignificant activities. This is the first demonstration that p300 has the capacity to catalyze Kbhb and the Kcr results are corroborated by published data (Lui X et al. Cell Discovery 2017 and Sabari BR et al. Mol Cell 2015) yet these data go unmentioned by the authors.

The observed differences between the enzymes make the generalized analysis performed in Figures 1c, 1d, and 1e unreasonable and obfuscating. Furthermore, the heavy-handed interpretation of these analyses seems inappropriate. It is surprising that the authors allow so much incongruity with their statements to go unmentioned in the text. Why generate these rich data matrices if they go unanalyzed?

We appreciate the comments by the reviewer and we agree that our conclusions were generalized. As the reviewer mentioned, this is a very complex subject and it was not our initial intent to describe the activity in detail of each individual enzyme given the large amount of data we generated. We considered, it was more useful to give the reader an overall sense of the results from our *in vitro* assays. Figure 1 was prepared using normalized data from the sum of all acylated peptides. We think the reviewer did an excellent analysis of our Supplementary Table, and we used it as initial step towards editing our manuscript. We hope the reviewer will appreciate our edits (Page 6).

...Previous studies have reported that the HATs CBP, p300 and PCAF can mediate propionylation¹², butyrylation^{12,31}, and crotonylation¹⁵ of lysine residues in vitro. These observations prompted us to investigate whether these and other known HATs can catalyze the acylation of human recombinant histones H3 and H4 using a broader range of acyl-CoA donors. We performed in vitro HAT activity assays with the HAT domains of PCAF, Gcn5, and the full-length CBP and p300 enzymes against histone H3, and with the HAT domains of MOF, Tip60 and NatA against histone H4. Each reaction was carried out individually in the

presence of eight different short-chain acyl-CoA donors, followed by bottom-up nano-LC-MS/MS analysis (Fig. 1a and 1b). Figure 1c summarizes the *in vitro* activity profiles of all HATs evaluated in this study. The heat map shows that most HATs could utilize acetyl-propionyl- and butyryl-CoA with relatively higher efficiency than the other acyl donors, supporting recent findings^{32,33}. However, acidic acyl-CoA donors including malonyl-, succinyl- and glutaryl-CoA, and branched-chain acyl-donors like β -hydroxybutyryl-CoA are utilized by HATs less efficiently. Interestingly, enzymes did not seem to utilize crotonyl-CoA for the catalysis of acyl-marks as effectively as propionyl- and butyryl-CoA despite the structural similarity within these cofactors. These data are in agreement with previous observations suggesting that HATs activity is weaker with crotonyl-CoA due to the planarity and rigidity imparted by the C-C double bond in the crotonyl moiety^{8,20,32}.

When taking a closer look at the individual acylation activities of all HATs (Supplementary Table 1), we observed that enzymes have different trends in their substrate preference. For instance, histone H3 is known to be selectively acylated at the lysine residue 14 (H3K14) by Gcn5³³ and PCAF³⁴. Therefore, if we look at the relative abundances for all acylations at H3K14, under our assay conditions, Gcn5 was able to butyrylate ~78% of the H3 peptide at position 14, whereas acetylation and propionylation were found at ~32% and ~11%, respectively (Supplementary Table 1). Likewise, PCAF displayed ~88% butyrylation, followed by ~5% crotonylation and ~2% acetylation at position H3K14. However, when looking at the average sum of the relative abundances of all acylated peptides by HATs on H3, there seems to be a trend in the order of substrate preference: acetyl > propionyl > butyryl > malonyl > succinyl > β -hydroxybutyryl > glutaryl > crotonyl (Fig. 1d, Supplementary Table 1). This trend inversely correlates with the increasing size of the side chain of the acyl donor (except for crotonyl-CoA and β -hydroxybutyryl), supporting the notion that the activity of HATs gets weaker with increasing acyl-chain length³². Interestingly, p300 and PCAF were the enzymes with the highest crotonylation activities on H3 (Supplementary Table 1).

Moreover, the average activities of HATs on histone H4 showed patterns that were consistent with the trend mentioned before in terms of substrate preference (Fig. 1d). However, individual acylation activities suggest that, while MOF seems to follow the same trend when looking at the sum of all acylated peptides, Tip60 prefers to utilize butyryl-CoA as a cofactor, followed by succinyl-CoA and acetyl-CoA (Supplementary Table 1). Nonetheless, all results reported in Figure 1 are based on the average contribution of both groups of HATs rather than individual acylation activities. Detailed acylation site specificities for all HATs can be found in Supplementary Table 1. Our data also showed that the N-terminal acetyltransferase NatA can catalyze N-terminal propionylation and butyrylation of histone H4 *in vitro* (Supplementary Fig. 2).

2. Figure 2:

The analysis of non-enzymatic acylation does not seem to take into account the 500uM concentration of acyl-CoAs used. As reviewer #2 pointed out, the rate of both enzymatic and non-enzymatic acylation will be dependent on substrate (acyl-CoA) concentration, but only the enzymatic reaction rate will saturate. By using excessive levels of acyl-CoA the authors are exaggerating the relative abundance of non-enzymatic acylation compared to enzymatic acylation. When trying to compare enzymatic versus non-enzymatic acylation, 500uM acyl-CoA will 1) underestimate the relative enzymatic contribution because these concentrations are much higher than measured Kms of HATs and most likely beyond concentrations for Vmax and 2) overestimate the non-enzymatic contribution as the rate of chemical acylation will not be throttled by an enzyme's Vmax. This is insufficiently acknowledged in the revision. The authors' citation of a study demonstrating that non-enzymatic acylation can also occur at physiological concentrations is irrelevant to these concerns.

Again, we agree with the reviewer. Although, we hope the reviewer agrees with us that this concentration of acyl-CoA was required to provide a comprehensive overview about the types and amount of acylation on histone peptides. Physiological amounts of acyl-CoA would make the statistics of this analysis prohibitive, because of their very low stoichiometry (as shown in the *in vivo* data). We have rewritten parts of the results of the non-enzymatic assay emphasizing that we are aware the concentrations used in the assay may result in an overestimation of the reactivity we are seeing *in vitro*. We hope the reviewer finds this version more appropriate (Page 8).

... The correlation between sites prone to chemical acylation (this study) and sites identified in other in vivo studies suggests that histone acyl modifications in cells could be the result of both enzymatic and non-enzymatic mechanisms after direct exposure to intrinsically reactive acyl-CoA metabolites. This is supported by the fact that enzymes with distinctive acyltransferase activities have not been identified in any cellular compartment⁴⁰. However, it is important to mention that concentrations of acyl-CoAs used in this experiment were far above the known physiological concentrations of CoA derivatives in whole cells⁴¹, so the extent of chemical acylation observed in this study is likely an overestimation. This higher concentration was required to ensure proper sensitivity to the in vitro assay. Even though it has been previously demonstrated that protein lysine acylation can occur at physiological acyl-CoA concentrations in vitro⁴², our study does not represent a suitable extrapolation for the reactivity of these intermediates in cells. As such, our data cannot rule out the possibility of the existence of enzymes that play a major role in catalyzing these marks in vivo as compared to non-enzymatic reactions.

“supporting the idea that most known HATs are unable to catalyze the acylation of histones using larger acyl-CoA donors”

The authors' own data demonstrate that several HATs are “able” to catalyze larger acylations. Instead of “unable,” the authors should write “HATs catalyze larger acylation less efficiently.” This is just one example of how the strength of the conclusions does not match the data presented. The authors should be more precise with their language.

We apologize for this issue. We have revised the mentioned text, and other similar statements in the rest of the manuscript, accordingly (Page 8).

...Again, we observed a trend in which the ratio of enzymatic/chemical reactivity of acyl groups is inversely correlated with the size of the side chain, supporting the idea that most known HATs catalyze larger acylations less efficiently^{32,33}

3. Figure 3

The reviewer appreciates that the authors redid the competition assays at a lower concentration of acyl-CoA, but it seems as though they misinterpreted the comments from the first revision. 10uM is still too high as it is well above the reported Km of most HATs and at the extreme end of reported whole cell concentrations. The authors' calling 10uM “low concentration of acyl-CoAs” does not make it so. Data from the literature demonstrate that crotonyl-CoAs can compete with acetyl-CoA for p300 acyltransferase activity at 2-4uM acyl-CoA concentrations (Sabari et al Mol Cell 2015).

Also, the data for the competition assays are not provided. Relevant mass spec data should be provided as a supplementary table.

We have revised the text accordingly and have also provided a supplementary table (Supplementary Table 3) with the relative abundances for individual peptides in the competition assay. Nevertheless, we would like to point out that 10 μM is not too far from the concentration of acyl-CoA we estimated in our work (Figure 4 and supplementary figure 5). Our data showed that acetyl, propionyl and succinyl-CoA are around 12, 1 and 0.5 μM , respectively. This is the reason why we selected this concentration for the repetition of the competition assay. (Page 9).

...To test this, we performed in vitro HAT competition assays in the presence of equimolar physiological concentrations of acetyl-CoA and other acyl-CoAs (Fig. 3a). As shown in figure 3b, most HATs preferred to utilize acetyl-CoA than any other acyl-CoA donor. It is important to mention that at 10 μM of acyl-CoAs, we observed non-enzymatic acylation of histones H3 and H4, as shown in the last two columns of the heatmap in figure 3b. Consistent with previous in vitro experiments, for most HATs, the preference for the competing cofactor, if any, largely depended on the size of the acyl donor side chain. As shown in figures 3c-3e, for the enzymes GCN5, p300 and MOF we observed an inverse correlation between the HAT preference for the competing cofactor and the increasing molecular weight of the acyl donor side chain, with the exception of crotonyl-CoA which was the least preferred cofactor by most HATs for the acylation of histones. Relative abundances for all peptides in competition assays are shown in Supplementary Table 3. All together, our data suggests that even in the highly unlikely chance that any other acyl-CoA accumulated to the extent that its concentration rivaled that of acetyl-CoA, HATs would still mostly utilize acetyl-CoA.

4. Figure 5

“However, the same correlation was not observed when performing the same analysis between sites that were non-enzymatically acylated in in vitro experiments with sites acylated during in nucleo experiments (Fig. S7).”

How does this not refute the relevance of your 500 μM acyl-CoA in vitro experiments or the statements made referencing them?

We have revised the mentioned text, and rewritten this section taken the reviewers comments into consideration (Page 13).

... MS analysis revealed that histone acylations can be induced in a concentration-dependent manner. Specifically, by adding 1 or 5 μM of acyl-CoAs we induced an increase of the respective acylation on histone peptides (Fig. 5b). Since the in nucleo experiment preserves the natural state of nuclear processes, it can be used to observe histone acylations in native chromatin. Our in vitro experiment demonstrated that acylations can occur by both enzymatic and non-enzymatic mechanisms, but such simplified assay cannot accurately represent the balances of a nuclear environment. We compared the two assays by performing an in nucleo – in vitro Spearman’s rank-order correlation analysis by using corrected in vitro enzymatic data (subtracting the non-enzymatic contribution). We observed a good correlation for some residues, including H3K9acyl and H3K18acyl sites (Supplementary Fig. 7) that were highly acylated only in presence of enzymes in vitro. This suggests that specific sites are likely more accessible to enzymatic activity than others, and that this reactivity is also a function of the acyl-CoA utilized. However, a generalized

conclusion cannot be drawn, as the in nucleo assay cannot discriminate enzymatic catalysis from chemical reaction, and physiological acylation turnover (equilibrium deposition/removal).

Mass spec data should be provided as a supplementary table.

5. Discussion

“However, they performed very poorly at catalyzing the acylation of histones with charged, branched or planar acyl-CoA cofactors.”

The authors’ own data does not support this statement (see comments on Figure 1). The authors’ should be more precise with their language.

We have revised this sentence following the reviewers suggestions (Page 15).

...However, they were less efficient catalyzing the acylation of histones with charged, branched or planar acyl-CoA cofactors

“our data does not support recent hypotheses suggesting that some novel acyl-PTMs could be mediated by any of the HATs explored in this study in vivo.”

The in vitro experiments, particularly the competition reactions, presented in this study have little relevance to the situation in vivo. In fact, the data from the authors’ in vitro and in nucleo experiments refute this statement. The authors show increases in sites of acylation for which they demonstrate HATs have the capacity to catalyze. The authors even make the statement about how the in nucleo acylation matches more to the enzymatic residues than to the non-enzymatic residues, which they define by using known HATs. Furthermore, some of the HATs utilize pr-CoA and but-CoA to equivalent levels as ac-CoA. Are they not included in “some novel acyl-PTMs”? The authors’ should be more precise with their language.

We appreciate the comments by the reviewer. We have removed this statement from the discussion.

“Thus, our data supports emerging models suggesting that non-enzymatic chemical reactions are a major contributor to the landscape of lysine acylations in nuclear histones (65).”

The only experiment that comes close to recapitulating the situation in a live cell, the in nucleo experiments, does not support this statement. The only cell-based experiment presented, myotube differentiation, shows a change in histone acylation that cannot easily be explained by non-enzymatic mechanisms. The only support for such a statement is the heavy-handed interpretation of single time-point, single concentration in vitro reactions. It is unclear where the authors are generating the confidence to make such a generalized statement about all histone acylations. The authors’ should be more precise with their language.

We agree with the reviewer that the *in nucleo* assay does not provide a direct evidence that increased histone acylation is due to non-enzymatic reaction. We observe a significant

increase in histone acylation when providing additional acyl-CoA to the nucleus (Figure 5b). Even though it is hard to believe that this minimal increase in, e.g. acetyl-CoA, makes enzymes so much more active, we indeed do not have data to support this is a non-enzymatic reaction. We have modified this paragraph (Page 15).

... Emerging hypotheses have suggested a model where non-enzymatic chemical reactions are a significant contributor to the landscape of lysine acylations in nuclear histones⁵⁶. They also suggest that sirtuin enzymes showing specificity for the removal of acyl marks may represent a constitutive programming to suppress potential damaging effects caused by the presence of these PTMs^{56,57}. Interestingly, our study showed that those succinylated and malonylated sites highly susceptible to non-enzymatic acylation in vitro were among the sites reported previously in in vivo studies³⁸. While more studies are required, our data suggest that histone lysine residues are prone to be modified by several free acyl-CoAs with and without enzymatic assistance.

Interestingly, a good in vitro – in nucleo correlation was observed for specific sites in histone H3, suggesting the implication of HATs in the utilization of supplemented acyl-CoAs for the acylation of histones.

How does this not refute your non-enzymatic claims? How can you ignore this discrepancy in logic?

We apologize for this sentence. This one indeed slipped away from our checks. We agree that it does not fit with our interpretations, and it is not supported by any data. It was removed.

6. Minor comments

Page 4:

1) “The dearth of quantitative data for these non-canonical acyl histone PTMs has led to the hypothesis that they might be nothing more than metabolic noise, arising simply from the chemical reactivity of acyl-CoAs.”

Loaded phrases like “nothing more than,” “noise,” “simply” don’t seem necessary. The same statement can be made as “led to the hypothesis that they might arise due to the chemical reactivity of acyl-CoAs.”

We have revised the text accordingly (Page 4).

...The dearth of quantitative data for these non-canonical acyl-PTMs has led to the hypothesis that they might arise due to the chemical reactivity of acyl-CoAs. Indeed, this has been observed in the context of acetylation and succinylation in mitochondrial proteins²⁸

2) “This last possibility might force us to reappraise their biological relevance”

This statement is unnecessarily charged and doesn’t seem appropriate to the scope of the paper.

We have removed this sentence from the introduction.

Page 8: “However, in competition assays perform in presence...” should be “performed”

We have corrected this typo.

Page 11: “shACL-treated cells” should be “shACLY-treated cells”

We have corrected this typo.

Page 16: “Intrigingly” should be “Intrigingly”

We have corrected this typo.

Page 17: “sugesting the implication of HATs in the utilization of supplemented...” should be “suggesting”

We have corrected this typo.

Page 33: Figure 5d. The colors used to represent data for H3K14pr and H3K14ac are too similar. As these are two colors the reader is trying to distinguish in the graph, it would be useful if they were more distinct from one another.

We have changed the colors for H3K14ac to dark purple.